# Quantification of absolute labeling efficiency at the single-protein level

Joschka Hellmeier [1,3], Sebastian Strauss [1,3], Shuhan Xu[1], Luciano A. Masullo [1], Eduard M. Unterauer[1], Rafal Kowalewski[1,2] & Ralf Jungmann [1,2] ✉

State-of-the-art super-resolution microscopy allows researchers to spatially resolve single proteins in dense clusters. However, accurate quantification of protein organization and stoichiometries requires a general method to evaluate absolute binder labeling efficiency, which is currently unavailable. Here we introduce a universally applicable approach that uses a reference tag fused to a target protein of interest. By attaching high-affinity binders, such as antibodies or nanobodies, to both the reference tag and the target protein, and then employing DNA-barcoded sequential super-resolution imaging, we can correlate the location of the reference tag with the target molecule binder. This approach facilitates the precise quantification of labeling efficiency at the single-protein level.

Optical fluorescence microscopy plays a pivotal role in modern biological and biomedical research, particularly in the investigation of macromolecular complexes within cells. Super-resolution microscopy has revolutionized the visualization of biological structures by overcoming the classical diffraction limit of light[1–4]. DNA points accumulation for imaging in nanoscale topography (DNA-PAINT)[5], which uses the transient binding of dye-labeled DNA oligos to their target-bound complements for super-resolution, enables unlimited multiplexing through DNA-barcoded sequential imaging (Exchange-PAINT)[6] as well as spatial resolution better than 5 nm at sufficiently high throughput[7–9]. Recent advancements in DNA-PAINT provide the technical capabilities to visualize biomolecules at true single-molecule resolution even in dense clusters[10]. However, the accurate quantification and interpretation of such datasets is limited by the inability to assess absolute labeling efficiency of binders used to label target molecules. Typically, a high-affinity binder such as a nanobody or antibody is used to label protein targets for super-resolution imaging[11]. However, the labeling process is not 100% efficient due to limited binder affinity, sterical hindrance or cell fixation artifacts.

Previous approaches to evaluate labeling efficiency employed diffraction-limited colocalization between target and reference samples[12,13] or have used nuclear pore complex proteins and their defined spatial arrangement as a reference standard[14]. Although this well-characterized protein complex is a promising approach, it is limited to the evaluation of binders against fusion tags (for example, monomeric enhanced green fluorescent protein (mEGFP) or ALFA-tag nanobodies) or nuclear pore proteins (for example, antibodies) and requires the generation of homozygous knock-in cell lines. In addition, DNA origami structures have been developed to assess the performance of labeling probes in vitro[15,16]. However, since these structures are not evaluated within a cellular context, the obtained results may not accurately reflect the labeling efficiency in a crowded or fixed cellular environment.

In this Article, to address these constraints, we present a widely adaptable method for assessing the binding efficiency of labeling probes at the single-molecule level within a cellular context. In brief, we present a molecular construct containing a reference tag attached to the protein of interest. Both the reference tag and the target protein are then labeled with a binder and subsequently imaged. Using both the reference and target channel, we can correlate the location of the reference tag with the target molecule binder, enabling precise quantification of labeling efficiency at the single-protein level. Our method is versatile and compatible with a range of super-resolution imaging techniques such as Stochastic Optical Reconstruction Microscopy (STORM)[4], Photoactivated Localization Microscopy (PALM)[3] or STimulated Emission Depletion (STED)[1], assuming adequate spatial resolution and single-molecule sensitivity for both reference and target binders is obtained. In general, the density of the target protein (adjustable by

[1]Max Planck Institute of Biochemistry, Planegg, Germany. [2]Faculty of Physics and Center for Nanoscience, Ludwig Maximilian University, Munich, Germany. [3]These authors contributed equally: Joschka Hellmeier, Sebastian Strauss. ✉e-mail: jungmann@biochem.mpg.de

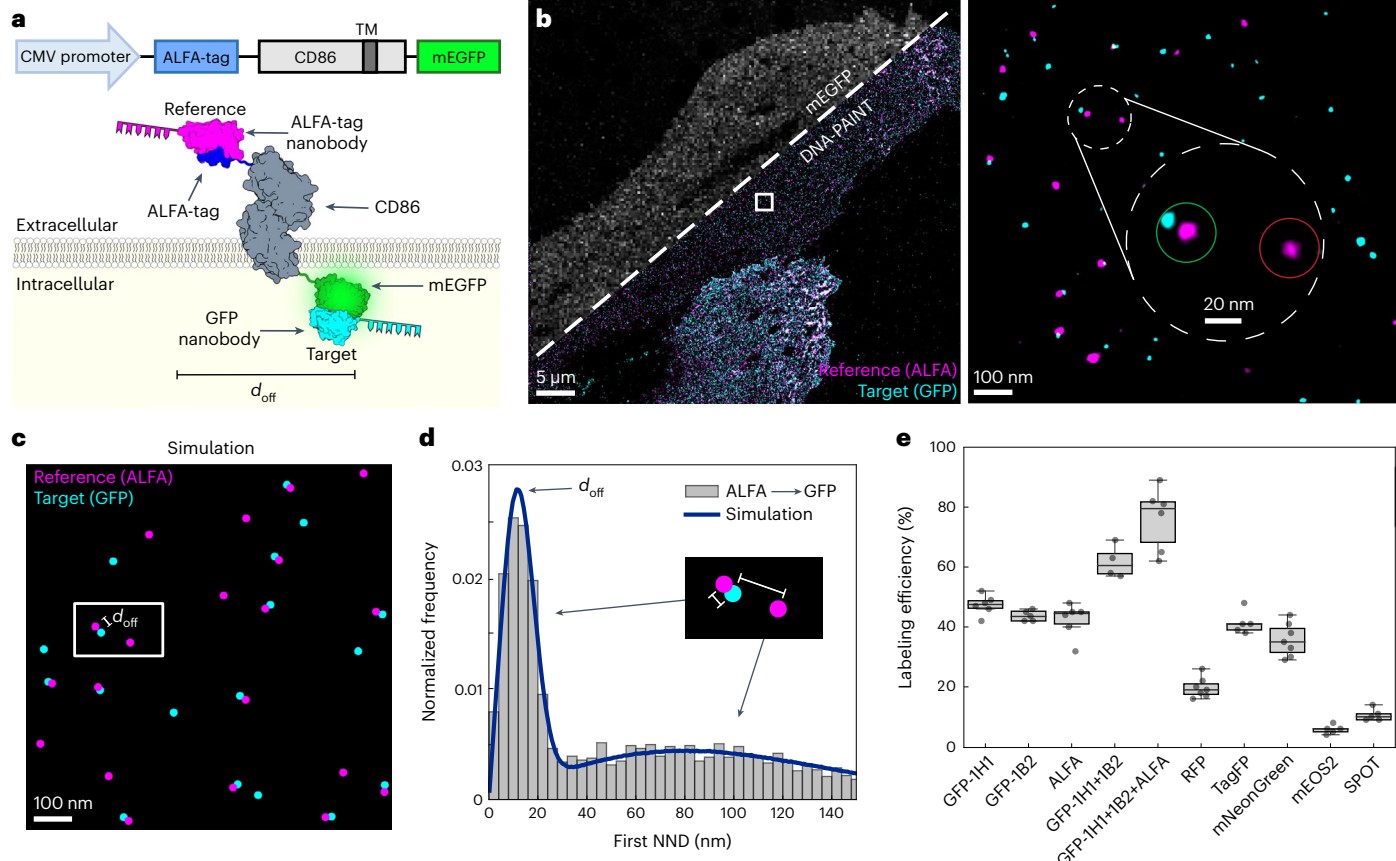

**Fig. 1 | Absolute quantification of labeling efficiency of tag proteins.**
**a**, Schematics of labeling efficiency analysis platform. For characterization of tag binders, CD86 served as a membrane anchor to localize target proteins to the intracellular side of the plasma membrane. ALFA-tag and tag of interest were inserted at the N- and C-terminus, respectively. Target and reference binders were separated by a distinct lateral offset distance doff. ALFA-tag nanobody (magenta) served as a reference to determine the labeling efficiency of target tag binder (for example, GFP-1H1 nanobody, cyan). **b**, Two-plex Exchange-PAINT image of transfected BSC1 ALFA−CD86−mEGFP cells. Zoom-ins depict super-resolution representations of target GFP-1H1 nanobody (cyan) colocalizing with ALFA-tag nanobody reference signal (magenta). Close-up view highlights colocalizing target and reference signal (green circle) and reference only signal (red circle). **c**, Illustration of corresponding two-population CSR simulation (brackets between the reference (magenta) and target (cyan)

represent the characteristic dimer distance $d_{off}$) **d**, Exemplary analysis of first NNDs of reference molecules to closest target signal. Fitting the NND data to a simulation model reveals the labeling efficiency of target binder. (The shorter bracket indicates the dimer distance $d_{off}$, while the longer bracket represents a random distance between two unrelated reference and targets molecules). **e**, Quantitative analysis as shown in **c** and **d** yielded the labeling efficiency of nanobodies against most widely used tags. Data are shown as a box plot, where the median is indicated by the center black line. Boxes extend from the 25th to the 75th percentile of each group's distribution of values. The whiskers extend to points that lie within 1.5 interquantile range of the lower and upper quartile. Data are of $n = 6$ (GFP-1H1 and ALFA-tag), $n = 7$ (mCherry and mNeonGreen) cells over three independent experiments and $n = 5$ (GFP-1B2, GFP-1H1, GFP-1B2, TagRFP, mEOS2 and SPOT-tag), $n = 6$ (GFP-1H1, GFP-1B2 and ALFA-tag) over two independent experiments.

selecting appropriate expression levels of the molecular construct) needs to match the spatial resolution capability of the employed imaging modality. Furthermore, single-protein sensitivity is necessary to accurately assess the target labeling efficiency. To fulfill these requirements, we used two-target Exchange-PAINT featuring sub-10 nm spatial resolution and dye-independent multiplexing.

In addition to offering researchers a procedure to evaluate the most suitable probes for their super-resolution imaging experiments, we furthermore want to emphasize the need for thorough evaluation of the target labeling efficiency. This is crucial for ensuring precise data interpretation and enabling reliable comparisons across different binders, labeling conditions and research laboratories.

## Results

### Workflow for absolute quantification of labeling efficiency
The basic procedure of our labeling efficiency evaluation approach is as follows: We design a molecular construct consisting of a reference tag and a target tag for which the labeling efficiency will be evaluated. After labeling with binders (for example, with antibodies), the sample

consists of the expressed constructs with either only the reference labeled, only the target labeled or both reference and target labeled. Independently of the labeling efficiency of the reference tag, the subset of constructs that contain a labeled reference can be used to quantify the labeling efficiency of a certain binder to the target in an absolute and quantitative manner. This is achieved by computing labeling efficiency = $N_{Ref+Target}/(N_{Ref} + N_{Ref+Target})$, that is, comparing the number of constructs that present both reference and target labeled ($N_{Ref+Target}$) to the total number of constructs that present the reference signal ($N_{Ref} + N_{Ref+Target}$).

Experimentally, we designed a construct consisting of CD86− a monomeric transmembrane protein[12]−for transient transfection incorporating an ALFA-tag as a reference at the N-terminus (that is, the ectodomain), and the target tag of interest at the C-terminus (that is, the endodomain) (Fig. 1a). We then record the location of the reference tag and target binders in a two-plex Exchange-PAINT experiment at single-protein resolution (Fig. 1b). At each reference binder position, we evaluate if a target binder is present in close proximity or not, accounting for random, density-based colocalization. For this purpose, we

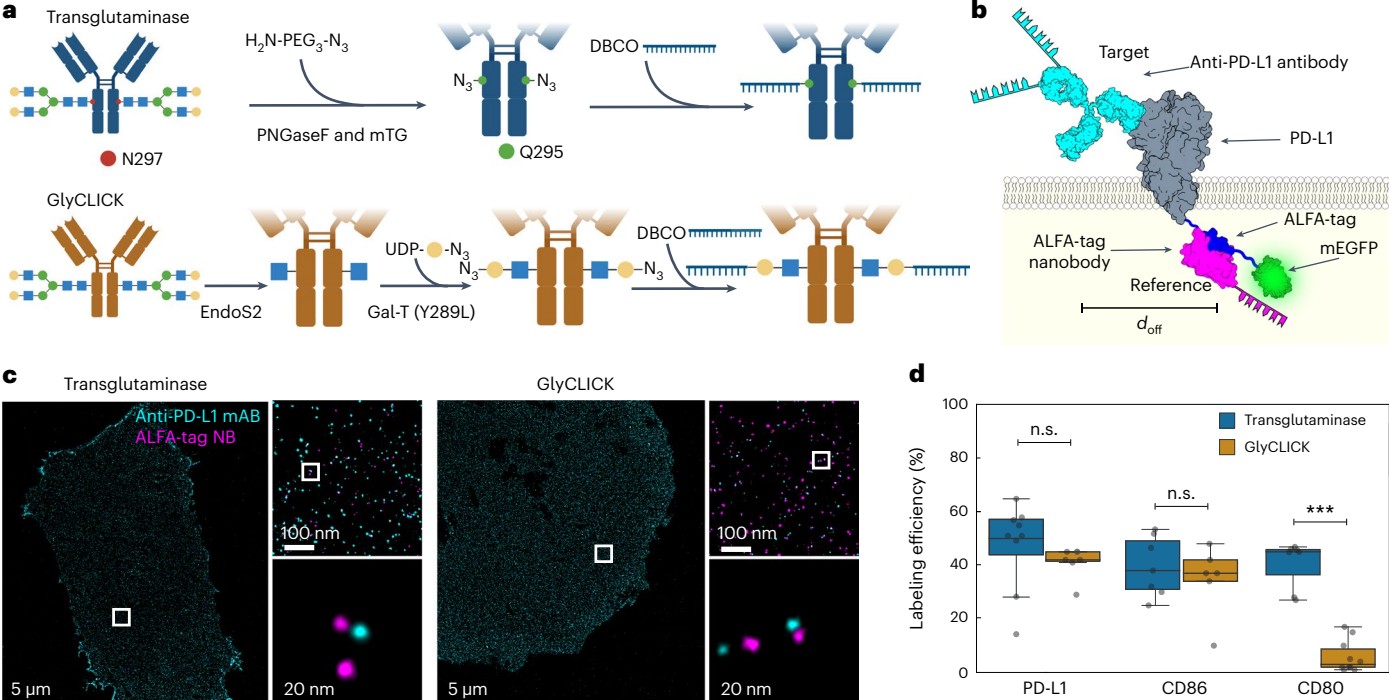

**Fig. 2 | Labeling efficiency evaluation of antibody conjugation methods.**
**a**, Schematics of antibody–DNA conjugation via PNGase/transglutaminase or GlyCLICK. For both, conjugation occurs at the $N$-glycan Fc part of the antibody yielding 1 or 2 DNA strands per antibody. **b**, Illustration of labeling efficiency determination for target anti-PD-L1 antibody (cyan). For antibody characterization, ALFA-tag was inserted at the PD-L1 C-terminus, while mEGFP served as an expression indicator. Colocalization of target and reference signal (magenta) revealed underlying labeling efficiency. **c**, Two-plex Exchange-PAINT image of CHO PD-L1–ALFA–mEGFP cells. Zoom-ins depict target anti-PD-L1 antibody colocalizing with respective ALFA-tag nanobody. **d**, Fitting the NND

data to a two-population model reveals the labeling efficiency of three different target antibodies. Data are shown as a box plot, where the median is indicated by the center black line. Boxes extend from the 25th to the 75th percentile of each group's distribution of values. The whiskers extend to points that lie within 1.5 interquantile range of the lower and upper quartile. Data are of $n = 9$ (PD-L1 and transglutaminase), $n = 7$ (PD-L1 and GlyCLICK, CD80 and transglutaminase, and CD86 and transglutaminase), $n = 10$ (CD80 and GlyCLICK) and $n = 5$ (CD86 and GlyCLICK) cells over three independent experiments and was tested for statistical significance via a two-sided bootstrap ratio test. ***$P < 0.001$, n.s., non-significant.

applied a cluster algorithm to identify individual molecules and their center of mass from clouds of localizations[10]. To accurately quantify labeling efficiency in an unbiased and automated way, we determine the cross nearest neighbor distance (NND) of each reference signal to its nearest target binder. We then simulate individual reference and target molecules as well as colocalizing reference and target molecules for the exact same measured experimental density (Fig. 1c and Extended Data Fig. 1a,d). Finally, we generate histograms of experimental and simulated NNDs and the most likely labeling efficiency is obtained through a least-squares minimization procedure (Fig. 1d, blue line, and Extended Data Fig. 1bc,e,f).

Using this approach, we quantified the labeling efficiencies of nanobody binders against widely used fusion tags, such as GFP, ALFA-tag, red fluorescent protein (RFP), TagRFP, mNeonGreen (mNG), mEOS2 and SPOT-tag. Interestingly, we observed substantial differences in labeling efficiencies of target nanobodies, ranging from almost 50% for anti-GFP (clone 1H1) to below 10% for anti-mEOS2 (clone 1E8) (Fig. 1e, Extended Data Fig. 2 and Supplementary Table 1). To further validate the GFP nanobody (clone 1H1) labeling efficiency, we exploited the well-characterized nuclear pore complex as a reference structure for direct counting of labeled Nup96–mEGFP subunits (Extended Data Fig. 3) yielding similar labeling efficiency values. Next, we investigated if labeling efficiency could be further increased by combining two nanobody clones (1H1 and 1B2) targeting two distinct GFP epitopes. Indeed, the labeling efficiency improved to 62 ± 5%. To evaluate if combining tags would lead to a further improvement in labeling efficiency, we designed a construct, where GFP and ALFA-tag are concatenated at the C-terminus of CD86 (Extended Data Fig. 2). We then used the two

GFP nanobody clones and the ALFA-tag nanobody as target binders, while employing a CD86 antibody as a reference probe. Strikingly, this resulted in a combined labeling efficiency of 76 ± 8%.

To rule out a potential influence of the membrane anchor protein as well as effects of membrane orientation of target and reference either on the extracellular or intracellular side or vice versa, first, we substituted CD86 with CD70 in the ALFA–CD86–mEGFP construct and second, swapped respective tag proteins. This resulted in a labeling efficiency for the GFP-1H1 nanobody not substantially different between CD86 and CD70 as membrane anchor proteins, regardless of the membrane orientation (Extended Data Fig. 4 and Supplementary Table 1).

## DNA-conjugation strategies affect underlying antibody labeling efficiency

In a next step, we extended our approach to evaluate the labeling efficiency of DNA-conjugated primary antibodies against the murine membrane proteins PD-L1, CD80 and CD86 to address potential effects of different labeling strategies. We employed two enzymatic, site-specific conjugation approaches targeting the Fc region of the primary antibodies. The first one modifies glutamines by deglycosylation with PNGase (peptide:$N$-glycosidase F) followed by targeting accessible glutamines with microbial transglutaminase. The second one comprises the hydrolysis of the Fc glycans to the inner most (N-acetylglucosamine (GlcNAc) moiety and second on the addition of the $N$-azidoacetylgalactosamine (GalNAz) to the inner most GlcNAc (GlyCLICK) (Fig. 2a). We designed the constructs with the target protein fused to a combined ALFA–mEGFP tag at the C-terminus, using mEGFP as a diffraction-limited transfection marker and the ALFA-tag nanobody

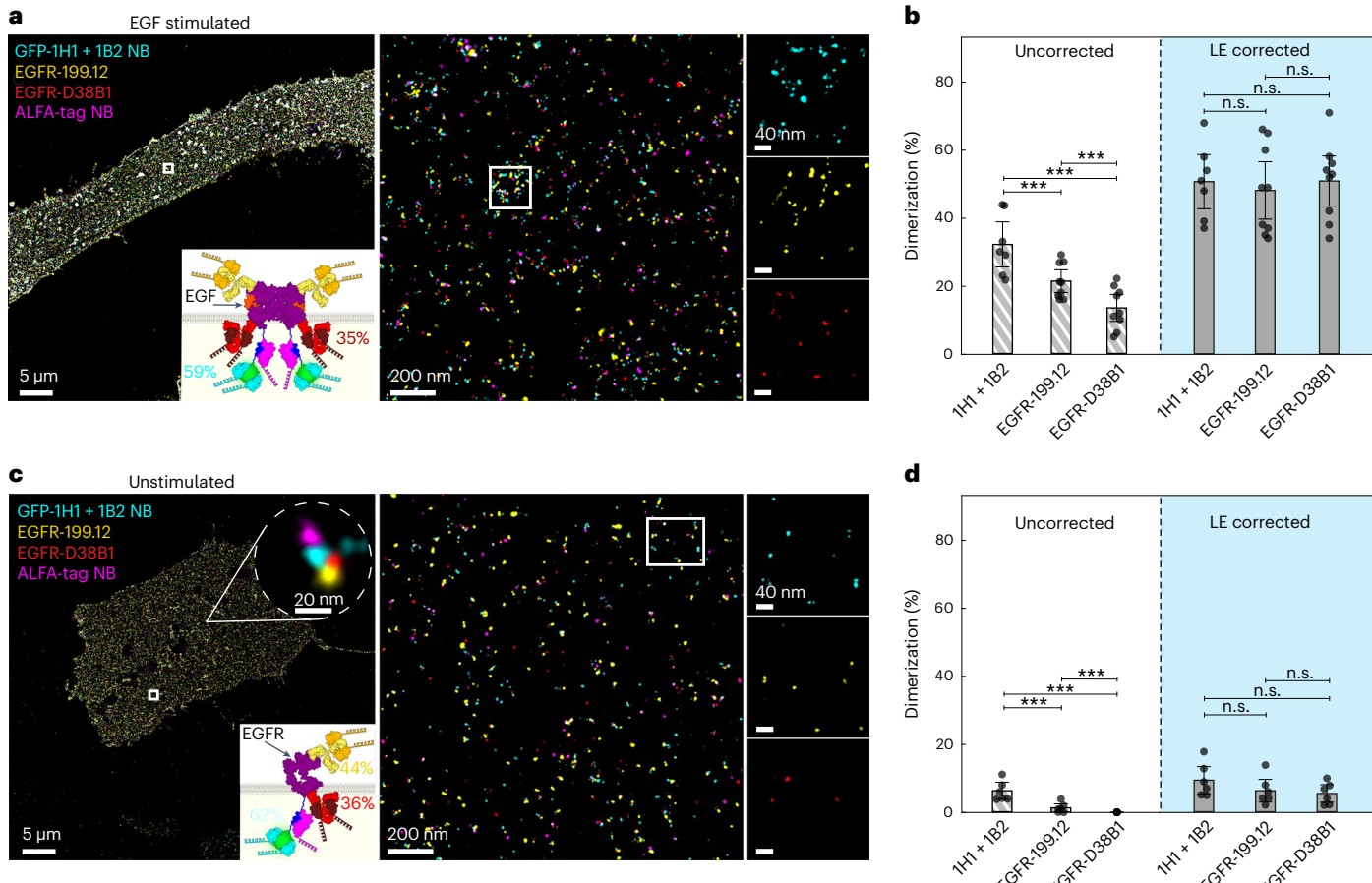

**Fig. 3 | Absolute quantification of EGFR dimerization. a**, Four-plex DNA-PAINT image of EGF-stimulated EGFR–ALFA–mEGFP transfected CHO cells. Three different binders (anti-GFP-1H1 + 1B2 nanobody (cyan), extracellular anti-EGFR antibody and anti-mouse kappa light chain nanobody (yellow) and intracellular anti-EGFR antibody and anti-rabbit IgG nanobody (red)) with different labeling efficiencies were used. Schematics of the construct for EGFR dimers is shown, with the respective mean labeling efficiency for each target binder (mean). Zoom-ins highlight the EGFR dimerization for each target binder. **b**, NND analysis yields quantitative information about the percentage of EGFR dimerization. For all target binders, EGFR dimerization was determined without and with accounting for labeling efficiency. **c**, Four-plex DNA-PAINT

image of unstimulated CHO EGFR–ALFA–mEGFP cells. Schematics of respective constructs together with corresponding labeling efficiencies (mean) for target binders. Zoom-ins depict spatial EGFR distributions for target binders in unstimulated cells. **d**, NND analysis yields quantitative information about the percentage of EGFR dimerization. For all target binders, EGFR dimerization was determined without and with accounting for labeling efficiency. Data are shown as the mean ± 95% confidence interval and of $n = 9$ (EGF stimulated) and $n = 6$ (nonstimulated) cells over three independent experiments. Data were tested for statistical significance via a two-sided bootstrap ratio test. ***$P < 0.001$; n.s., non-significant; LE, labeling efficiency.

as a reference tag binder (Fig. 2b, and Extended Data Figs. 5 and 6). As described before, we record the location of the reference tag and target binders in a two-plex Exchange-PAINT experiment at single-protein resolution (see Fig. 2c for PD-L1 exemplary case). We then compared the absolute labeling efficiency for PD-L1, CD80 and CD86 antibodies, which were DNA-conjugated using either transglutaminase or Gly-CLICK strategies. While we obtained similar labeling efficiencies for both labeling approaches in the case of the anti-PD-L1 and anti-CD86 antibodies, there was a stark difference between transglutaminase and GlyCLICK in the case of the anti-CD80 antibody, with an almost sevenfold lower efficiency for the latter conjugation approach (Fig. 2d and Supplementary Table 2).

**Absolute quantification of EGFR dimerization is biased by underlying binder labeling efficiency**

Finally, we aimed to demonstrate the importance of absolute quantification of labeling efficiencies for the investigation of the oligomerization of the epidermal growth factor receptor (EGFR), which dimerizes upon EGF stimulation[17,18]. To this end, we transiently expressed EGFR–ALFA–mEGFP on the surface of Chinese hamster ovary (CHO) cells

and evaluated labeling efficiencies of multiple different EGFR binders (primary antibodies EGFR-199.12 and EGFR-D38B1, the affibody ab95116, and GFP nanobodies 1H1 + 1B2 GFP; Extended Data Fig. 7 and Supplementary Table 3). Based on the performance results, we chose the two primary antibodies for direct EGFR labeling and GFP nanobodies (clones 1H1 + 1B2) targeting the mEGFP tag, with labeling efficiencies of 49 ± 4%, 35 ± 9% and 59 ± 4%, respectively. While the existence of preformed EGFR dimers in the unstimulated case is still under debate[19,20], we expect a certain percentage of dimerized EGFR proteins in the EGF-stimulated scenario[17,18]. Based on the different absolute labeling efficiencies of the antibodies and GFP nanobodies, we hypothesize an apparent difference in the absolute EGFR dimerization percentage among the binders. We in fact observed that in the stimulated case (Fig. 3a and see also Extended Data Fig. 8 for validation that the labeling efficiency was consistent for unstimulated and EGF-stimulated cells), each binder resulted in substantially different EGFR dimer fractions (ranging from 13% for the least efficient EGFR binder to 32% for the GFP nanobody), correlating with their respective labeling efficiency. We then used the determined labeling efficiencies from above and calculated a labeling efficiency-adjusted dimerization

fraction obtaining similar dimerization fractions (~50%) independent of the used binders (Fig. 3b and Supplementary Table 4). This highlights the importance of absolute quantification of labeling efficiency for accurate data interpretation. Next, we performed the same experiment, analysis and labeling efficiency correction for the unstimulated case. Similar to the stimulated case, we measured notable variations of apparent EGFR dimer contributions ranging from 0% for the least efficient EGFR binder to 6% for the GFP nanobody labeling strategy. Only after correcting for the labeling efficiency, we obtained an EGFR dimer contribution of 7% for all binders (Fig. 3c,d and Supplementary Table 4), again highlighting the importance of absolute quantification of labeling efficiency.

## Discussion

In conclusion, we have developed a method to evaluate absolute labeling efficiency of binders, which is essential to accurately quantify for example protein amounts, organization and interactions using super-resolution microscopy. Our approach is generally applicable to a large variety of target molecules, tags and probes. We have demonstrated this for various tag nanobodies and primary antibodies using membrane proteins as example target molecules. Our method is straightforward to implement, only requiring cloning and transient transfection of constructs containing the target proteins fused to reference tags. For the first time, we are offering comprehensive guidance to researchers regarding the highest performing tags and their corresponding binders. We show that tags can be concatenated and binders combined to boost labeling efficiency beyond 80%. In addition to raw binder performance, we show that different DNA-conjugation methods to label primary antibodies can considerably influence labeling efficiency, underlining the necessity of thorough binder characterization to enable truthful data interpretation. Finally, taking labeling efficiency into quantitative consideration, we provide an approach to account for differences in labeling efficiency. This allowed us to achieve consistent results for the dimerization percentages of EGFR proteins post-EGF stimulation independent of the employed labeling probes.

We note, that it is important to conduct our workflow for evaluating labeling probe efficiency under the same cell fixation and target staining conditions that will be applied in subsequent experiments, where these probes are employed for protein target quantification. This is paramount as differences in fixation, permeabilization and staining protocols will probably affect the resulting absolute probe labeling efficiency. Our assay can be customized to match specific experimental conditions, allowing for the direct benchmarking of probe labeling efficiency in the real-world conditions of downstream experiments. Finally, we want to emphasize again that our presented labeling efficiency assay can be performed beyond a DNA-PAINT readout. Our approach is adaptable to various super-resolution imaging methods, potentially including STORM, PALM and STED, provided that both reference and target binders can be visualized with the necessary spatial resolution and single-molecule sensitivity. It is crucial to match the target protein density (which can be adjusted by choosing suitable expression levels for the tagged protein construct) to the spatial resolution capabilities of the chosen imaging technique. The sparsity of target molecules is particularly important for a potential extension to live cells. In this case, our approach could consist of essentially dual-target single-molecule tracking experiments using an external labeling probe to be evaluated in conjunction with an intracellular self-labeling tag (for example, SNAP or Halo tag) as reference, which in turn could be labeled with cell-permeable dyes.

Looking ahead, systematic studies can now be performed, evaluating effects such as fixation conditions, conjugation methods, DNA sequences or additional modifications. Our approach to quantify absolute labeling efficiencies together with recent advancements in DNA-based imaging[10] opens the door to absolutely quantitatively assess protein amount, organization and interaction on the level of single biomolecules, paving the way for quantitative spatial proteomics[21].

## Online content

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

## Methods

### Materials

C3-azide and Cy3B modified DNA oligonucleotides were ordered from Metabion and MWG Eurofins. Ultrapure water (cat. 10977-035), Tris 1 M pH 8 (cat. AM9855G), EDTA 0.5 M pH 8.0 (cat. AM9260G) and 10× phosphated-buffered saline (PBS) pH 7.4 (cat. 70011051) were purchased from Thermo Fisher Scientific. Sodium chloride 5 M, bovine serum albumin (cat. A9647) and sodium azide (cat. 71289) were ordered from Sigma-Aldrich. The 16% formaldehyde methanol-free (cat. 28908) was obtained from Thermo Fisher Scientific. Triton X-100 (10% solution) (cat. 93443), Tween 20 (cat. P9416-50ML) and glycerol (cat. 65516-500ML), protocatechuate 3,4-dioxygenase pseudomonas (PCD; cat. P8279), 3,4-dihydroxybenzoic acid (PCA; cat. 37580-25G-F) and (±)-6-hydroxy-2,5,7,8-tetra-methylchromane-2-carboxylic acid (Trolox; cat. 238813-5G) were ordered from Sigma-Aldrich. Fetal bovine serum (cat. 10500-064), 1× PBS pH 7.2 (cat. 20012-019), 0.05% Trypsin–EDTA (cat. 25300-054) and Lipofectamine 3000 (cat. L3000015) were purchased from Thermo Fisher Scientific. The 90 nm diameter gold nanoparticles (cat. G-90-100) were ordered from Cytodiagnostics. μ-Slide eight-well high glass bottom (cat. 80807) was purchased from ibidi. Amicon Ultra-0.5 and Amicon Ultra-2 centrifugal filter units with 10 kDa and 50 kDa molecular weight cutoff (MWCO; cat. UFC5010, UFC 5050, UFC201024 and UFC205024) were purchased from Merck.

### Buffers

The following buffers were used for sample preparation and imaging:

- Imaging buffer: 1× PBS, 1 mM EDTA, 500 mM NaCl, pH 7.4, supplemented with 1× Trolox, 1× PCA and 1× PCD
- Washing buffer: 1× PBS, 1 mM EDTA and 0.02% Tween 20
- Blocking buffer: 1× PBS pH 7.4, 1 mM EDTA, 2% bovine serum albumin and 0.02% Tween 20.

### Trolox, PCA and PCD

The 100× Trolox consisted of 100 mg Trolox, 430 μl 100% methanol and 345 μl 1 M NaOH in 3.2 ml $H_2O$. The 40× PCA consisted of 154 mg PCA, 10 ml water and NaOH, which were mixed together. The pH was adjusted 9.0. The 100× PCD consisted of 9.3 mg PCD, 13.3 ml of buffer (100 mM Tris–HCl pH 8, 50 mM KCl, 1 mM EDTA and 50% glycerol).

### Plasmid cloning

gBlocks for the described inserts were ordered from IDT and individually cloned into a plasmid cloning DNA 3.1 (+) Mammalian Expression Vector (Thermo Fisher Scientific, cat. V79020).

### GlyCLICK antibody–DNA conjugation

Up to four batches of 500 μg unconjugated antibodies were buffer exchanged to Tris-buffered saline (TBS) and concentrated to 100–200 μl by using Amicon centrifugal filters (50 kDa MWCO). Antibodies were deglycosylated using immobilized GlycINATOR microspin columns (Genovis, cat. A0-GL6-050) according to the manufacturer's instructions. First, the columns were equilibrated three times with 300 μl TBS each time and centrifuged at 200g for 1 min. For deglycosylation antibodies were added to the column and incubated at room temperature for 30 min with end-over-end mixing. Afterwards, antibodies were collected by centrifugation at 1,000g for 1 min. To maximize the recovery, 100 μl TBS was added to the column and centrifuged at 1,000g for 1 min. In a second step, the deglycosylated antibodies were functionalized with an azide using the Genovis azide activation kit (Genovis, cat. L1-AZ1-200). A total of 7 μl buffer additive was added per 550 μl deglycosylated antibody solution. UDP-GalNAz was reconstituted in 40 μl TBS and 10 μl was added per antibody. Finally, 10 μl beta-1,4-galactosyltransferase (Y289L) was added per antibody and the reaction mix was incubated at 30 °C overnight. The next day, enzymes and excess UDP-GalNAz were removed using Amicon centrifugal filters

(50 kDa MWCO); then tenfold molar excess of dibenzocyclooctyne (DBCO)-DNA was added and incubated overnight in the shaker at 25 °C at 300 rpm. Nonreacted DNA was removed by buffer exchange to PBS (pH 7.2) using Amicon centrifugal filters (50 kDa MWCO), and subsequently the unconjugated antibody was removed by anion exchange chromatography using an ÄKTA pure system equipped with a Resource Q 1 ml column and antibody concentration was adjusted to 5 μM. DNA-conjugated antibodies were stored at 4 °C.

### Transglutaminase antibody–DNA conjugation

Antibody–DNA conjugation via peptide-N-glycosidase F (PNGase) and microbial transglutaminase was performed as published previously[22]. Before functionalization, unconjugated antibodies (PD-L1, CD80 and CD86) were concentrated to 1 mg ml$^{-1}$ in TBS and 0.05% Tween20 by using Amicon centrifugal filters (50 kDa MWCO). For each conjugation, 100 μg of the respective antibody was used. A total of 0.6 U PNGase (Roche, cat. 11365193001), 1.2 U microbial transglutaminase (Zedira, cat. T001) and an 80-fold molar excess of bifunctional azide-PEG3-amine linker (Jena Bioscience, cat. CLK-AZ101-100) were added to the antibody and reacted on a shaker for 16 h at 37 °C, 300 rpm. Enzymes and excess linker were removed by buffer exchange to PBS using Amicon centrifugal filters (50 kDa MWCO). Azide-modified antibodies were reacted with 10× molar excess of DBCO-functionalized DNA (R1–CD86, R3–CD80 and R4–PD-L1) overnight at 25 °C, 300 rpm. Nonreacted DNA was removed by buffer exchange to PBS (pH 7.2) using Amicon centrifugal filters (50 kDa MWCO). Antibody concentration was adjusted to 5 μM and stored at 4 °C.

### Nanobody–DNA conjugation via single cysteine

Nanobodies against GFP, tagFP, and rabbit and mouse IgG were purchased from Nanotag with a single ectopic cysteine at the C-terminus for site-specific and quantitative conjugation. The conjugation to DNA-PAINT docking sites (Supplementary Table 5) was performed as described previously[23]. First, buffer was exchanged to 1× PBS + 5 mM EDTA, pH 7.0, using Amicon centrifugal filters (10 kDa MWCO) and free cysteines were reacted with 20-fold molar excess of bifunctional maleimide-DBCO linker (Sigma-Aldrich, cat. 760668) for 2–3 h on ice. Unreacted linker was removed by buffer exchange to PBS using Amicon centrifugal filters. Azide-functionalized DNA was added with 3–5 M excess to the DBCO-nanobody and reacted overnight at 4 °C. Unconjugated nanobody and free azide DNA was removed by anion exchange using an ÄKTA Pure liquid chromatography system equipped with a Resource Q 1 ml column. Nanobody–DNA concentration was adjusted to 5 μM (in 1× PBS, 50% glycerol and 0.05% NaN$_3$) and stored at −20 °C.

### Nanobody–DNA conjugation via sortase-tag

The nanobody against SPOT-tag was purchased with a sortase recognition motif LPETG (Proteintech, cat. etb-250). The 0.5 M equivalents of calcium-independent sortase 7 M (obtained from MPI Core Facility) and 5 M equivalents of Gly-Gly-Gly-DNA (Biomers) were added to 200 μg nanobody and reacted at 37 °C for 45 min. Nanobody–DNA conjugates were purified by anion exchange using an ÄKTA Pure liquid chromatography system equipped with a Resource Q 1 ml column. Nanobody–DNA concentration was adjusted to 5 μM (in 1× PBS, 50% glycerol and 0.05% NaN$_3$) and stored at −20 °C.

### Cell culture

BSC1 (ATCC, CCL-26) and CHO-K1 (ATCC, CCL-61) cells were cultured in Dulbecco's modified Eagle medium (Thermo Fisher Scientific, cat. 16600082) and F-12K medium (Thermo Fisher Scientific, cat. 21127030), respectively, and supplemented with 10% fetal bovine serum.

### Binder specificity

A total of 20,000 BSC1 or CHO-K1 cells per well were seeded and allowed to adhere overnight. The 4% formaldehyde solution was preheated to

37 °C before addition to the cells. Cells were fixed in 4% formaldehyde for 15 min and washed with PBS. Cells were permeabilized in 0.125% Triton X-100 in PBS for 5 min and washed with PBS, followed by passivation with blocking buffer for 60 min at room temperature. DNA-conjugated binders were diluted in blocking buffer and added at a final concentration of 50 nM (target, Supplementary Table 6) and 500 pM (reference, Supplementary Table 6) for 60 min at room temperature. Unbound binders were removed by washing with washing buffer followed by washing once with Imaging buffer for 5 min. Post-fixation was performed with 2% formaldehyde in PBS for 5 min. Before the addition of gold fiducials, samples were washed with PBS. Subsequently, 250 µl of 90 nm standard gold nanoparticles, diluted 1:3 in PBS, were added and incubated for 5 min before washing with PBS.

## Tag-protein labeling efficiency
BSC1 cells were seeded on eight-well high glass-bottom chambers 1 day before transfection at a density of 10,000 cells per well. BSC1 cells were transfected with a single receptor construct (ALFA–CD86–mEGFP (GFP-1H1, GFP-1B2 and GFP-1H1 + 1B2), ALFA–CD86–mEOS, ALFA–CD86–mNeonGreen, ALFA–CD86–TagRFP, ALFA–CD86–SPOT, mEGFP–CD86–ALFA (GFP-1H1), ALFA–CD70–mEGFP (GFP-1H1), mEGFP–CD70–ALFA (GFP-1H1)) at a time for binder characterization using Lipofectamine 3000 as specified by the manufacturer. Cells were allowed to express receptors for 16–24 h. 4% formaldehyde solution was preheated to 37 °C before addition to the cells. Cells were fixed in 4% formaldehyde for 15 min and washed with PBS. Cells were permeabilized in 0.125% Triton X-100 in PBS for 5 min, washed with PBS followed by blocking in blocking buffer for 60 min at room temperature. DNA-conjugated nanobody (target, Supplementary Table 7) and ALFA-tag nanobody (reference, Supplementary Table 7) were diluted in blocking buffer and added at a final concentration of 50 nM and 500 pM for 60 min at room temperature, respectively. Unbound binders were removed by washing with washing buffer, followed by washing once with imaging buffer for 5 min. Post-fixation was performed with 4% formaldehyde in PBS for 10 min. Cells were washed with PBS, and subsequently 250 µl of 90 nm standard gold nanoparticles, diluted 1:3 in PBS, were added and incubated for 5 min, followed by washing with PBS.

## Nup96–mEGFP labeling efficiency
U2OS–CRISPR–Nup96–mEGFP cells were cultured in ibidi eight-well high glass-bottom chambers at a density of 30,000 cells per well. The 2.4% formaldehyde solution was preheated to 37 °C before addition to the cells. The cells were fixed with 2.4% paraformaldehyde in PBS for 30 min. After fixation, the cells were washed with PBS. Subsequently, 90 nm standard gold nanoparticles, diluted 1:3 in PBS, were incubated for 5 min and then washed with PBS. Blocking and permeabilization were carried out with 0.25% Triton X-100 in blocking buffer for 90 min at room temperature. After another round of washing with PBS, the cells were incubated with 50 nM anti-GFP nanobodies (clone 1H1) in blocking buffer for 60 min at room temperature. Unbound binders were removed by washing with PBS, followed by a single wash with imaging buffer for 10 min. Post-fixation was performed with 2.4% paraformaldehyde in PBS for 15 min followed by washing with PBS.

## Antibody labeling efficiency
CHO cells were seeded on eight-well high glass-bottom chambers 1 day before transfection at a density of 10,000 cells per well. CHO cells were transfected with a single receptor construct (PD-L1–ALFA–mEGFP, CD80–ALFA–mEGFP, CD86–ALFA–mEGFP) at a time for binder characterization using Lipofectamine LTX as specified by the manufacturer. CHO cells were allowed to express receptors for 1 day. The 4% formaldehyde solution was preheated to 37 °C before addition to the cells. Cells were fixed in 4% formaldehyde for 15 min and washed with PBS. Cells were permeabilized in 0.125% Triton X-100 dissolved in PBS for 5 min, washed with PBS followed by blocking for 60 min at room temperature.

DNA-conjugated antibody (target, Supplementary Table 7) was diluted in blocking buffer and added at a final concentration of 50 nM overnight at 4 °C. The next morning ALFA-tag nanobody (reference, Supplementary Table 7) was dissolved in blocking buffer at a final concentration of 500 pM and added for 60 min at 24 °C. Unbound binders were removed by washing with washing buffer, followed by washing once with imaging buffer for 5 min. Post-fixation was performed with 4% formaldehyde in PBS for 10 min. Cells were washed with PBS, and subsequently 250 µl of 90 nm standard gold nanoparticles, diluted 1:3 in PBS, were added and incubated for 5 min, followed by washing with PBS.

## EGFR labeling efficiency
CHO cells were seeded in eight-well high glass-bottom chambers 1 day before transfection at a density of 10,000 cells cm$^{-2}$. CHO cells were transfected with a single receptor construct (EGFR–ALFA–mEGFP) at a time for binder characterization using Lipofectamine LTX as specified by the manufacturer. CHO cells were allowed to express receptors for 16–24 h. The 4% formaldehyde solution was preheated to 37 °C before addition to the cells. Cells were fixed in 4% formaldehyde for 15 min and washed with PBS. Cells were permeabilized in 0.125% Triton X-100 in PBS for 5 min, washed with PBS and followed by passivation with blocking buffer for 60 min at room temperature.

**GFP nanobodies.** DNA-conjugated GFP nanobodies (clone 1H1, clone 1B2) (target, Supplementary Table 7) and ALFA-tag nanobody (reference, Supplementary Table 7) were diluted in blocking buffer and added at a final concentration of 50 nM and 500 pM for 60 min at room temperature, respectively. Unbound binders were removed by washing with washing buffer, followed by washing once with imaging buffer for 5 min. Post-fixation was performed with 4% formaldehyde in PBS for 5 min. Before the addition of gold fiducials, samples were washed with PBS. Subsequently, 250 µl of 90 nm standard gold nanoparticles, diluted 1:3 in PBS, was added and incubated for 5 min before washing with PBS.

**Affibody.** DNA-conjugated affibody (target, Supplementary Table 7) and ALFA-tag nanobody (reference, Supplementary Table 7) was diluted in blocking buffer and added at a final concentration of 50 nM and 500 pM for 60 min at room temperature, respectively. Unbound binders were removed by washing with washing buffer, followed by washing once with imaging buffer for 5 min. Post-fixation was performed with 2% formaldehyde in PBS for 5 min. Before the addition of gold fiducials, samples were washed with PBS. Subsequently, 250 µl of 90 nm standard gold nanoparticles, diluted 1:3 in PBS, were added and incubated for 5 min before washing with PBS.

**Antibody.** DNA-conjugated antibody (target, Supplementary Table 7) was diluted in blocking buffer and added at a final concentration of 50 nM overnight at 4 °C, followed by washing with washing buffer. The next morning, the secondary nanobody and ALFA-tag nanobody (reference, Supplementary Table 7) were diluted in blocking buffer at a final concentration of 50 nM and 500 pM, respectively, and added for 60 min at room temperature. Unbound binders were removed by washing with washing buffer, followed by washing once with imaging buffer for 5 min. Post-fixation was performed with 2% formaldehyde in PBS for 5 min. Before the addition of gold fiducials, samples were washed with PBS. Subsequently, 250 µl of 90 nm standard gold nanoparticles, diluted 1:3 in PBS, were added and incubated for 5 min before washing with PBS.

## EGF-stimulation EGFR
CHO cells were seeded on eight-well high glass-bottom chambers the day before transfection at a density of 15,000 cells per well. CHO cells were transfected with EGFR–ALFA–mEGFP using Lipofectamine 3000 as specified by the manufacturer. CHO cells were allowed to

express receptors overnight. Then, CHO cells were starved by incubating them in serum-free F-12K medium for 6 h. EGF was diluted in PBS at a final concentration of 10 nM and preheated to 37 °C before addition to the cells. Cells were then stimulated for 10 min at 37 °C. Unstimulated CHO cells served as a reference. Next, 4% formaldehyde was preheated to 37 °C and cells were fixed for 15 min and washed with PBS. Cells were permeabilized in 0.125% Triton X-100 in PBS for 5 min and washed with PBS followed by incubation with blocking buffer for 12 h at 4 °C. In a next step, anti-EGFR antibodies (EGFR-199.12 and EGFR-D38B1, Supplementary Table 8) were diluted 1 in 100 in blocking buffer and incubated overnight at 4 °C, followed by washing with washing buffer. The next morning, anti-GFP nanobodies (clone 1H1, clone 1B2, Target, Supplementary Table 8), secondary nanobody (mouse kLC 1A23 and rabbit IgG 10E10, target, Supplementary Table 8) together with ALFA-tag nanobody (reference, Supplementary Table 8) were diluted in blocking buffer at a final concentration of 50 nM and 500 pM, respectively, and added for 60 min at room temperature. Unbound binders were removed by washing with washing buffer, followed by washing with imaging buffer for 5 min. Post-fixation was performed with 2% formaldehyde in PBS for 5 min at room temperature. Before the addition of gold fiducials, samples were washed with washing buffer. Subsequently, 250 µl of 90 nm standard gold nanoparticles, diluted 1:3 in PBS, were added and incubated for 5 min before washing with PBS.

### DNA-PAINT imaging for binder specificity determination

Imaging of wild-type BSC1 or CHO-K1 cells was conducted via imaging single target binders using distinct imagers for each binder (Supplementary Table 9). Cy3B-conjugated imager strands were added at a final concentration of 1 nM in imaging buffer and 600 µl of the imager solution was added to the sample to perform DNA-PAINT measurements. Imaging parameters are listed in detail in Supplementary Table 6. Unspecific binding of target binders is given in detail in Supplementary Table 10.

### DNA-PAINT imaging of Nup96–mEGFP

Imaging of U2OS–CRISPR–Nup96–mEGFP cells was conducted via imaging single target binders (GFP nanobody, clone 1H1) using R3 as imager strand. Cy3B-conjugated imager strands were added at a final concentration of 500 pM in imaging buffer and 600 µl of the imager solution was added to the sample to perform a standard DNA-PAINT measurement on the sample. A detailed overview of imaging parameters is listed in Supplementary Table 7.

### Two-plex Exchange-PAINT imaging for binder labeling efficiency determination

First, transfected cells were identified through detection of the fluorescent protein signal (for example, mEGFP, mCherry and tagRFP). Next, imager strands (Supplementary Table 9) specific to the target binder docking strands were introduced and a DNA-PAINT image was recorded for all selected cells. Afterwards, imager strands were washed off with PBS and imager strands complementary to the reference binder were added followed by imaging. A detailed overview of imaging parameters is listed in Supplementary Table 7.

### Four-plex Exchange-PAINT imaging for EGFR oligomerization determination

EGFR–ALFA–mEGFP positive CHO-K1 cells were identified through detection of the fluorescent mEGFP signal. Multiplexed cellular imaging was conducted via four subsequent imaging rounds using four different imager strand sequences (Supplementary Table 9) with only one of the imagers present at a time. Cy3B-conjugated imager strands were diluted in imaging buffer and 600 µl of the imager solution was added to the sample to perform DNA-PAINT measurements. In between imaging rounds, the sample was washed with 2 ml PBS until no residual

signal from the previous imager solution was detected followed by addition of the next imager solution. A detailed overview of imaging parameters is listed in Supplementary Table 8.

### Microscope setup

Fluorescence imaging was carried out on an inverted microscope (Nikon Instruments, Eclipse Ti2) with the Perfect Focus System, applying an objective-type total internal reflection fluorescence (TIRF) configuration equipped with an oil-immersion objective (Nikon Instruments, Apo SR TIRF ×100, numerical aperture of 1.49, oil). A 560 nm laser (MPB Communications, 1 W) was used for excitation and coupled into the microscope via a Nikon manual TIRF module. The laser beam was passed through a cleanup filter (Chroma Technology, ZET561/10) and coupled into the microscope objective using a beam splitter (Chroma Technology, ZT561rdc). Fluorescence was spectrally filtered with an emission filter (Chroma Technology, ET600/50 m and ET575lp) and imaged on an sCMOS camera (Andor, Zyla 4.2 Plus) without further magnification, resulting in an effective pixel size of 130 nm (after 2 × 2 binning). TIR illumination was used. The readout rate of the camera was set to 540 MHz. Images were acquired by choosing a region of interest with a size of 512 × 512 pixels. Raw microscopy data were acquired using µManager[24] (Version 2.0.1).

### Image analysis

Raw fluorescence data were subjected to super-resolution reconstruction using the Picasso software package[25] (latest version available at ref. [26]). Drift correction was performed with a redundant cross-correlation and gold particles as fiducials for cellular experiments. Gold particles were also used to align all rounds for two-plex Exchange-PAINT experiments. After channel alignment, DNA-PAINT data were analyzed using the Picasso clustering algorithm (latest version available at ref. [26]) for each target individually. Circular clusters of localizations centered around local maxima were identified and grouped (assigned a unique identification number). Subsequently, the centers of the localization groups were calculated as weighted mean by employing the squared inverse localization precisions as weights.

### Data analysis of binder specificity

Binder specificity was evaluated by counting the number of target binder signals (circular clusters of localizations centered around local maxima) in wild-type BSC1 or CHO-K1 cells and determining underlying binder density within the cell area.

### Data analysis of labeling efficiency

To quantify the labeling efficiency of a given binder the NND distribution extracted from the data is compared to a simulation. Briefly, the simulation consists of simulating monomers of the reference protein, monomers of the target protein and dimers of reference-target protein at different proportions. Subsequently, the NND distribution of a given simulation is calculated and compared to the experimental NND distribution. The most likely proportions of populations of monomers ($p_{Ref}$) and dimers ($p_{Ref+Target}$) were obtained through a least-squares optimization procedure. The labeling efficiency is then calculated as labeling efficiency percentage $= p_{Ref+Target}/(p_{Ref} + p_{Ref+Target}) \times 100$.

The algorithm of the simulation can be summarized as follows:

- Parameters. Density of target monomers: number of target monomers per unit area. Density of reference monomers: number of reference monomers per unit area. Density of target-reference dimers: number of dimers per unit area. Dimer distance: expected distance between reference and target molecule including the labeling construct. Uncertainty: variability in the position of each molecule due to labeling and localization errors. The total density for target and reference is set to match the respective experimentally observed densities in each channel.

- Simulation of monomers: a set of spatial coordinates with complete spatial randomness (CSR) distribution and given density are drawn. Simulation of dimers: a set of spatial coordinates with CSR distribution are drawn, representing the center of each dimer. For each dimer center two positions are generated with a random orientation and a distance with expected value dimer distance. The position of each pair of molecules are drawn taking into account the uncertainty parameter (drawn from a Gaussian distribution).
- NND are calculated on the subset of detectable molecules.

### Data analysis of EGFR dimerization

To quantify the spatial interactions of the EGFR, NND-based analysis was performed for unstimulated and EGF-stimulated cells using the same region of interest for the three different EGFR datasets (GFP nanobodies (clone 1H1 and clone 1B2), EGFR 199.12 and EGFR D38B1). First, the NND of each single EGFR was computed followed by comparison of the experimental histogram of NND to numerical simulations of combinations of populations of oligomers as described below. The most likely proportions of populations of oligomers were obtained through a least-squares optimization procedure. CSR distributions of monomer and dimer populations were simulated and corresponding NNDs calculated.

The algorithm of the simulation can be summarized as follows:

- Parameters. Density of monomers: number of monomers per unit area. Density of dimers: number of dimers per unit area. Dimer distance: expected distance between the two molecules including the labeling construct. Uncertainty: variability in the position of each molecule due to labeling and localization errors. Labeling efficiency: fraction of ground-truth molecules that will actually be labeled and measured. The observed density, which has to match the experimental parameter, then becomes observed density = (density of monomers + density of dimers) × labeling efficiency.
- Simulation of monomers: a set of spatial coordinates with CSR distribution and given density are drawn. Simulation of dimers: a set of spatial coordinates with CSR distribution are drawn, representing the center of each dimer. For each dimer center two positions are generated with a random orientation and a distance with expected value of dimer distance. The position of each pair of molecules are drawn taking into account the uncertainty parameter (drawn from a Gaussian distribution).
- A random subset of 'detectable' molecules is taken from the ground-truth set (fraction = labeling efficiency) to simulate the labeling process.
- NND are calculated on the subset of detectable molecules.

#### Reporting summary

Further information on research design is available in the Nature Portfolio Reporting Summary linked to this article.

## Data availability

Localization data from this study are available via Zenodo at https://doi.org/10.5281/zenodo.10718926 (ref. 27). Raw microscopy data obtained during this study are available from the corresponding author on reasonable request. All raw data are available upon reasonable request from the authors. Source data are provided with this paper.

## Code availability

Raw image processing can be performed using Picasso available via GitHub at https://github.com/jungmannlab/picasso (ref. 26) with documentation available via Picasso at https://picassosr.readthedocs.io/en/latest/render.html (ref. 28). Custom software to calculate labeling efficiency is available via Zenodo at https://doi.org/10.5281/zenodo.107189277 (ref. 27).

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

## Acknowledgements

This research was funded in part by the European Research Council through an ERC Consolidator Grant (ReceptorPAINT, grant agreement number 101003275), the BMBF (Project IMAGINE, FKZ: 13N15990), the Volkswagen Foundation through the initiative 'Life?—A Fresh Scientific Approach to the Basic Principles of Life' (grant no. 98198), the Max Planck Foundation and the Max Planck Society. S.X. acknowledges a PhD fellowship from the Boehringer Ingelheim Fonds and support by the IMPRS-LS graduate school. L.A.M. acknowledges a postdoctoral fellowship from the European Union's Horizon 20212022 research and innovation program under Marie Skłodowska-Curie grant agreement no. 101065980. Molecular graphics were generated with UCSF ChimeraX, developed by the Resource for Biocomputing, Visualization and Informatics at the University of California, San Francisco, with support from National Institutes of Health R01-GM129325 and the Office of Cyber Infrastructure and Computational Biology, National Institute of Allergy and Infectious Diseases.

## Author contributions

J.H. and S.S. conceived and performed experiments. S.X. designed plasmids for transfection. L.A.M. designed and wrote the analysis software. J.H., L.A.M., E.M.U. and R.K. analyzed the data. J.H., S.S., S.X. and L.A.M. contributed to the writing of the manuscript. R.J. conceived and supervised the study, interpreted data and wrote the manuscript. J.H. and S.S. contributed equally. All authors reviewed and approved the final manuscript.

## Funding

## Competing interests

J.H., S.S. and R.J. have filed a patent on the method published in this paper. The other authors declare no competing interests.

## Additional information

**Extended data** is available for this paper at https://doi.org/10.1038/s41592-024-02242-5.

**Correspondence and requests for materials** should be addressed to Ralf Jungmann.

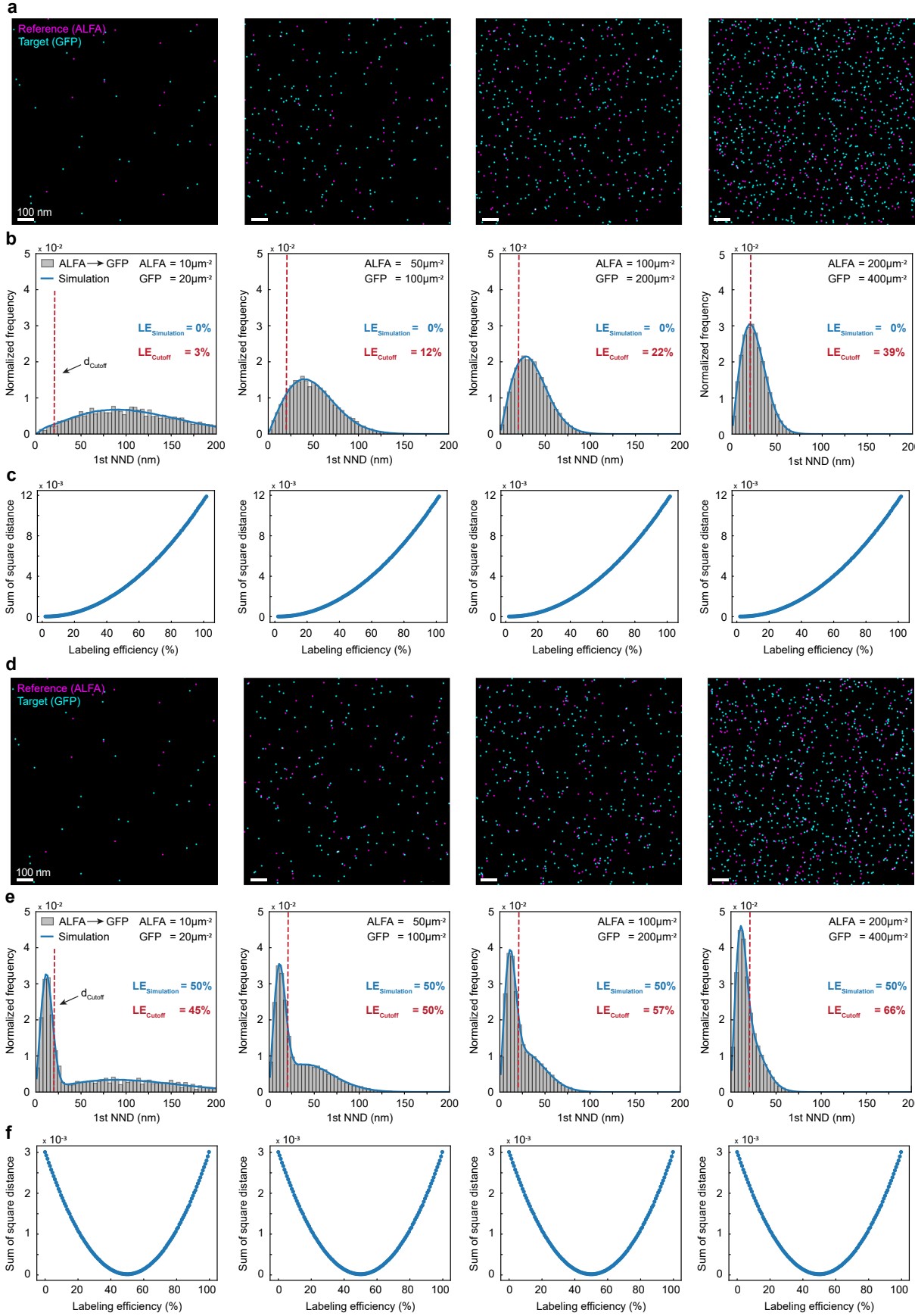

**Extended Data Fig. 1 | See next page for caption.**

**Extended Data Fig. 1 | Unbiased determination of labeling efficiency.**
Simulated 2-plex Exchange-PAINT image of ALFA-tag nanobody (=Reference, magenta) and GFP nanobody (=Target, cyan) signals at a given labeling efficiency (LE) of either **a**, 0% or **d**, 50%. For testing robustness of determined labeling efficiency values, we generated data sets of randomly distributed molecule positions on a $5\,\mu m \times 5\,\mu m$ grid at multiple different densities (ALFA = $10\,\mu m^{-2}$, $50\,\mu m^{-2}$, $100\,\mu m^{-2}$, $200\,\mu m^{-2}$; GFP = $20\,\mu m^{-2}$, $100\,\mu m^{-2}$, $200\,\mu m^{-2}$, $400\,\mu m^{-2}$). Labeling efficiency is quantified in an unbiased and automated way by determination of cross-nearest neighbor distances (NND) of each reference signal to its nearest target signal (**b**,**e**). Individual reference and target molecules as well as colocalizing reference and target molecules for the underlying density were simulated and histograms were generated of all NNDs. The most likely LE is obtained through a least-squares minimization procedure where the sum of the square distances between experimental and simulated data are computed. Blue dots in **c** and **f** represent the sum of the square distances for different values of LE, the most likely LE is the one that minimizes such sum, resulting in LE = 0% and L = 50% respectively. Alternatively, LE is determined based on a cutoff distance of 20 nm (red dashed line) which was experimentally determined for ALFA-tag nanobody and GFP nanobody ($d_{offset}$ = 10 nm, $2\sigma$ = 10 nm). Note that using a cutoff produces inaccurate results and especially overestimates te LE at higher molecular densities.

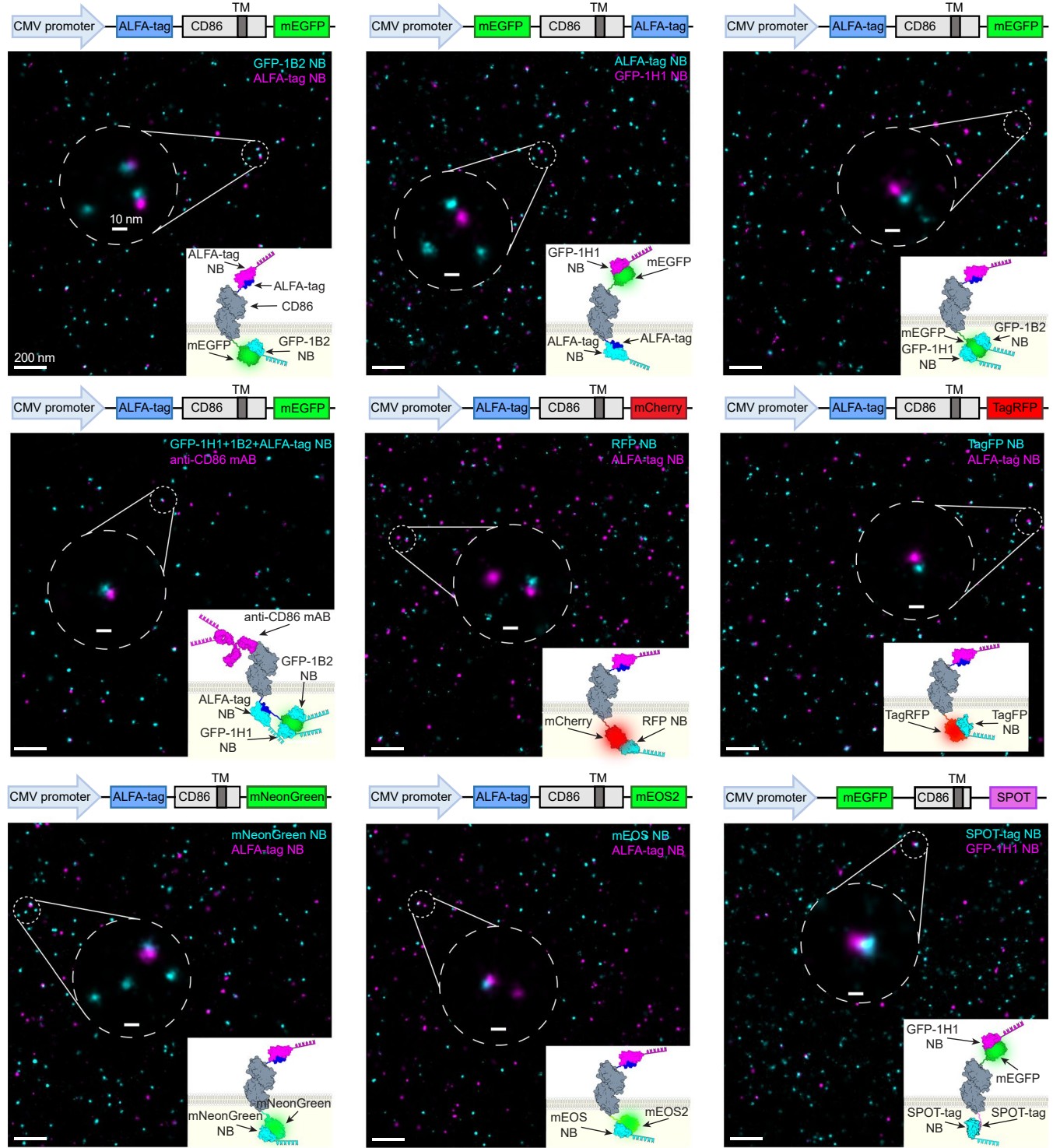

**Extended Data Fig. 2 | Labeling efficiency determination of intracellular tag binders.** 2-plex Exchange-PAINT image of transiently transfected BSC1 cells expressing the monomeric membrane receptor CD86. For characterization of intracellular tag binders targeting mEGFP, ALFA-tag, mCherry, TagRFP, mNeonGreen, mEOS2 or SPOT-tag, the intracellular tag of interest was inserted at the CD86 C-terminus and the reference tag (for example ALFA-tag, mEGFP) was inserted at the CD86 N-terminus on the extracellular side of the cell surface. To notably boost the labeling efficiency against a single target protein, For mEGFP, nanobody clones were combined as well as ALFA-tag and mEGFP were concatenated at theCD86 C-terminus to boost labeling efficiency. For concatenated tag, anti-CD86 antibody served as a reference. A fluorescent tag (for example mEGFP, mCherry) served as an indicator for construct expression. Zoom-ins depict super-resolution representation of target binder (cyan) colocalizing with respective reference binder signal (magenta).

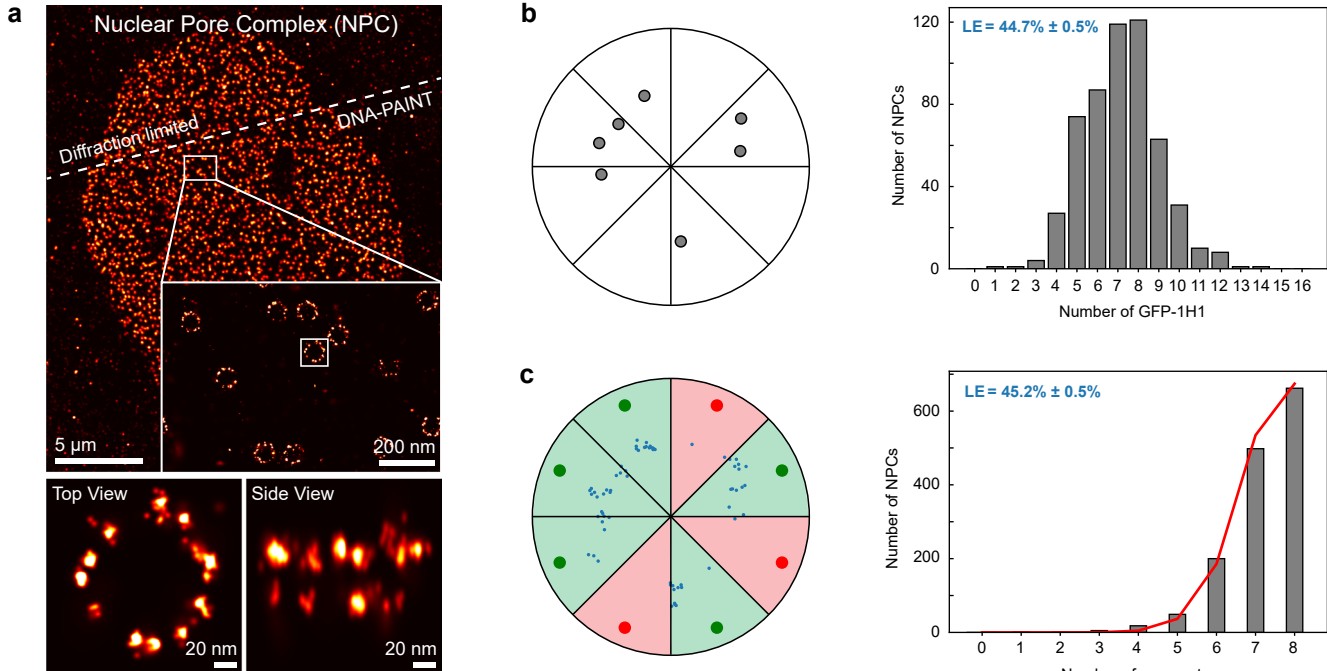

**Extended Data Fig. 3 | Labeling efficiency determination of Nuclear Pore Complex (NPC) proteins. a**, Diffraction-limited and DNA-PAINT overview image of Nup96-mEGFP labeled with DNA-conjugated anti-GFP-nanobodies (clone 1H1). Zoom-in illustrates the 8-fold symmetry of the NPC. Individual GFP positions were resolved at single protein resolution. Nup96 is present in 32 copies per NPC, 16 copies in the nuclear and cytoplasmic ring (NR and CR, respectively). **b**, NPCs were filtered based on their median localization precision (≤3 nm), aligned by their centers of mass, NR and CR were separated based on the z-coordinates of localizations and GFP-1H1 positions were determined based on respective 2D projection using Picasso's 'SMLM clusterer'. Labeling efficiency (LE) of GFP-1H1 was determined by evaluating the average number of GFP-1H1 molecules per NPC ring (7.15 ± 0.07) divided by the 16 possible GFP-1H1 positions for each ring, yielding an overall LE of 44.7% ± 0.5%. **c**, For comparison, LE was determined via NPC linking and filtering of localizations, alignment and segmentation of the NPCs. Underlying LE was determined by fitting the histogram of numbers of segments that contain at least 4 localizations to a probabilistic model, yielding LE of 45.2% ± 0.5%. Uncertainties in the reported LE values were obtained by bootstrapping with 20 re-sampled data sets. Data is of 2 cells over 2 independent experiments.

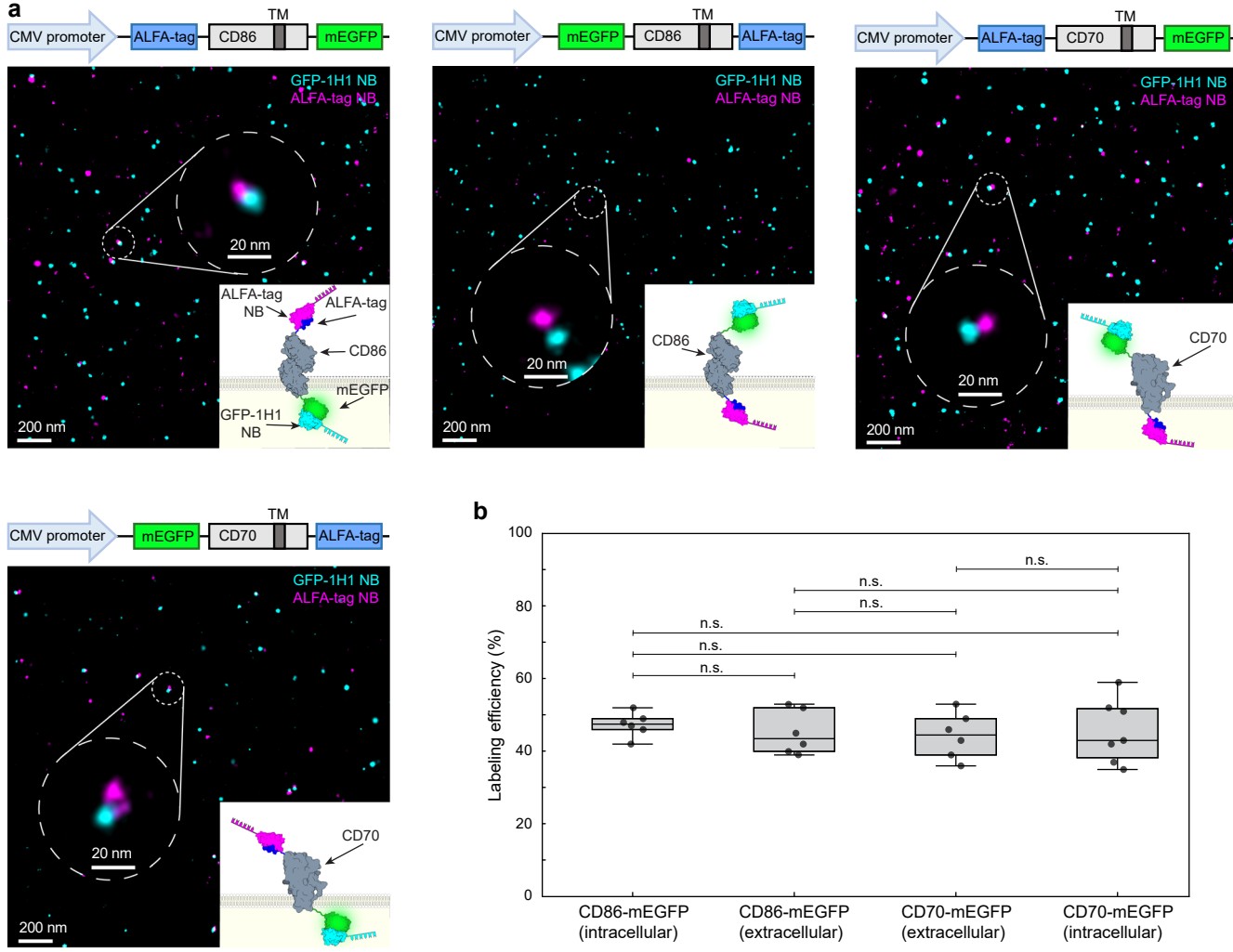

**Extended Data Fig. 4 | Labeling efficiency of mEGFP binder for different membrane proteins intracellular vs. extracellular. a**, 2-plex Exchange-PAINT image of transiently transfected BSC1 cells expressing either ALFA-CD86-mEGFP, mEGFP-CD86-ALFA, ALFA-CD70-mEGFP or mEGFP-CD70-ALFA on the cell surface. mEGFP was inserted either at the C-terminus of the membrane receptor CD86 (mEGFP = intracellular) and CD70 (mEGFP = extracellular) or the N-terminus of CD86 (mEGFP = extracellular) and CD70 (mEGFP = intracellular). ALFA-tag was inserted at the respective N- or C-terminus of corresponding membrane receptors. For determination of underlying labeling efficiency of target tag binder (cyan) ALFA-tag nanobody (magenta) served as a reference. mEGFP served as an indicator for construct expression. Zoom-ins depict super-resolution representation of target GFP nanobody (clone 1H1) (cyan) colocalizing with respective reference ALFA-tag nanobody (magenta). Corresponding offset distance is highlighted in the close-up view. **b**, Quantitative analysis of first nearest neighbor distances (NND) of target-reference signal and fitting the NND data to a two-population model reveals underlying labeling efficiency. Data is shown as a box plot, where the median is indicated by the center black line. Boxes extend from the 25th to the 75th percentile of each group's distribution of values. The whiskers extend to points that lie within 1.5 interquartile range of the lower and upper quartile. Data is of n = 6 (CD86-GFP – C-terminal, CD86-GFP – N-terminal, CD70-GFP – N-terminal) and n = 7 (CD70 – C-terminal) cells over 3 independent experiments. Data was tested for statistical significance via a two-sided bootstrap ratio test. n.s., non-significant.

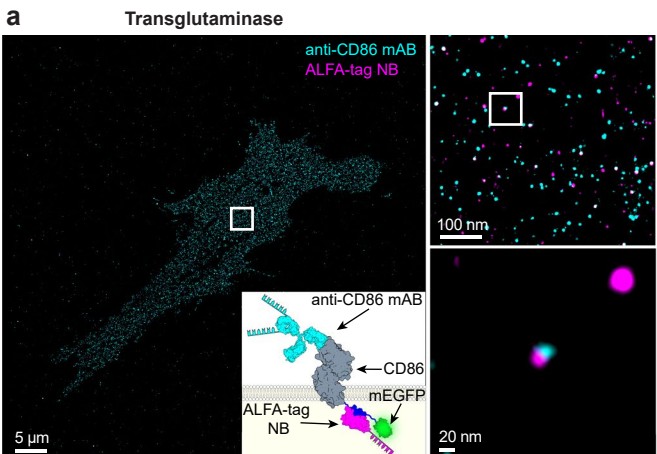

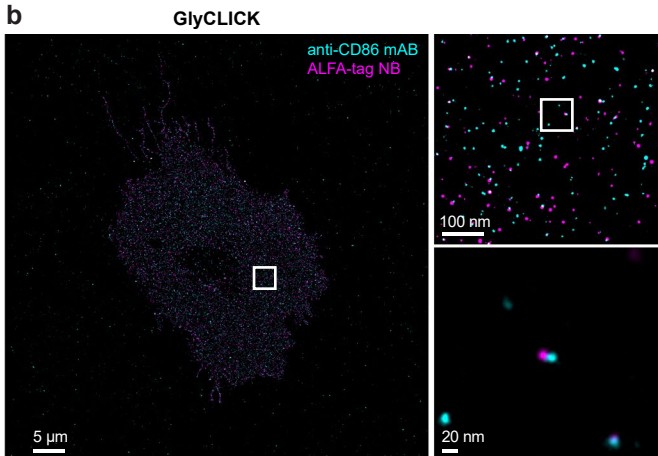

**Extended Data Fig. 5 | Labeling efficiency evaluation of site-specific anti-CD86 antibody conjugation.** 2-plex Exchange-PAINT image of transiently transfected CHO cells expressing CD86-ALFA-mEGFP on the cell surface. Respective anti-CD86 antibody was site-specifically modified via either transglutaminase **a**, or via GlyCLICK **b**, DNA-strand conjugation. Schematic sketch of labeling efficiency determination for target anti-CD86 antibody (cyan) specifically binding the membrane receptor CD86. For antibody characterization ALFA-tag was inserted at the CD86 N-terminus on the intracellular side of the cell surface and the fluorescent protein mEGFP served as an indicator for surface expression. Colocalization of target antibody and ALFA-tag nanobody reference signal (magenta) revealed underlying labeling efficiency. Zoom-ins depict super-resolution representation of target anti-CD86 antibody colocalizing with respective ALFA-tag nanobody.

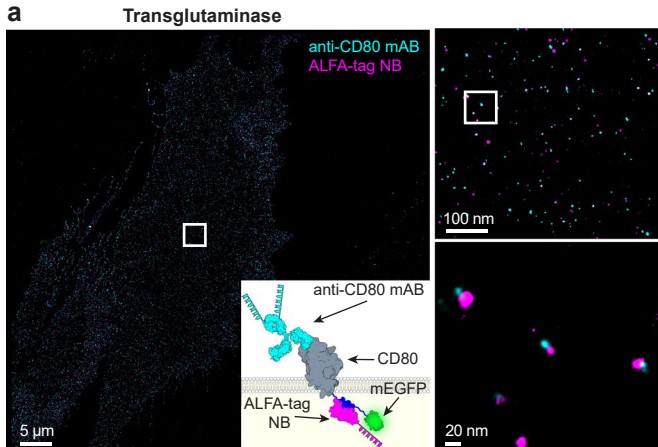

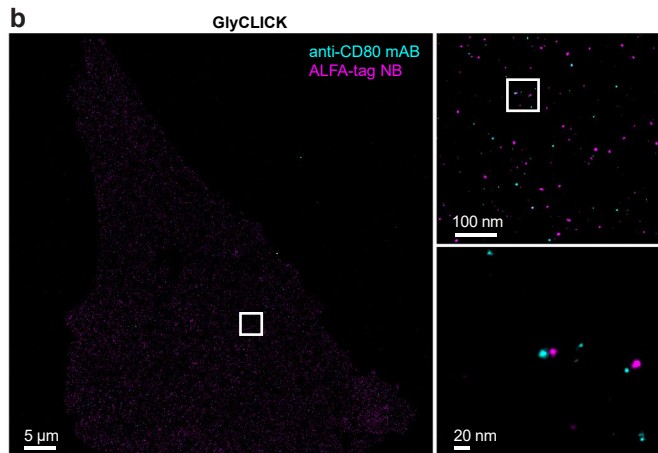

**Extended Data Fig. 6 | Labeling efficiency evaluation of site-specific anti-CD80 antibody conjugation.** 2-plex Exchange-PAINT image of transiently transfected CHO cells expressing CD80-ALFA-mEGFP on the cell surface. Respective anti-CD80 antibody was site-specifically modified via either transglutaminase **a**, or via GlyCLICK **b**, DNA-strand conjugation. Schematic sketch of labeling efficiency determination for target anti-CD80 antibody (cyan) specifically binding the membrane receptor CD80. For antibody characterization ALFA-tag was inserted at the CD80 N-terminus on the intracellular side of the cell surface and the fluorescent protein mEGFP served as an indicator for surface expression. Colocalization of target antibody and ALFA-tag nanobody reference signal (magenta) revealed underlying labeling efficiency. Zoom-ins depict super-resolution representation of target anti-CD80 antibody colocalizing with respective ALFA-tag nanobody.

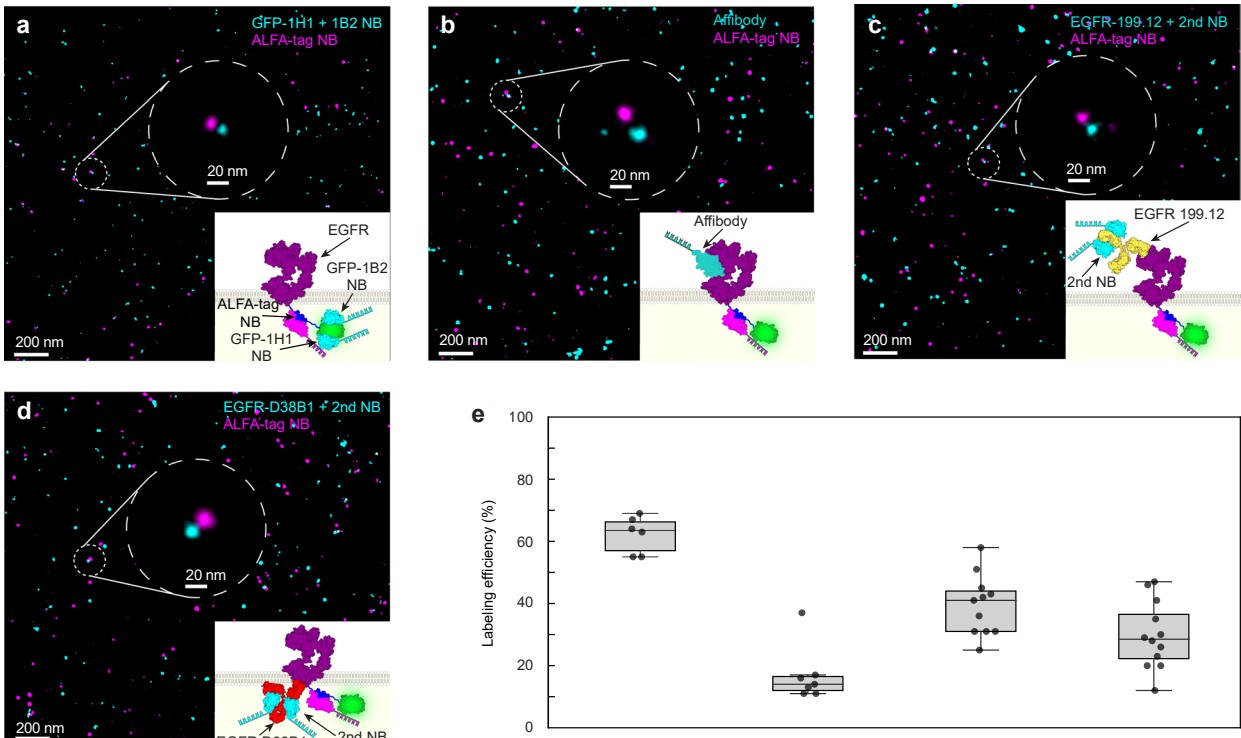

**Extended Data Fig. 7 | Determination of underlying labeling efficiency of EGFR binders.** 2-plex Exchange-PAINT image of transiently transfected CHO cells expressing EGFR-ALFA-mEGFP on the cell surface. EGFR molecules were labeled either via two GFP-nanobodies binding two different epitopes **a**, via an EGFR affibody **b**, via an extracellular anti-EGFR antibody labeled at the Fc part via 2[ary] nanobodies **c**, or via an intracellular anti-EGFR antibody labeled at its kappa-chain via 2[ary] nanobodies (**d**). Schematic sketches of labeling efficiency determination for respective EGFR binders (cyan) are illustrated. For binder characterization ALFA-tag was inserted at the EGFR N-terminus on the intracellular side of the cell surface and the fluorescent protein mEGFP served as an indicator for successful receptor surface expression. Colocalization of target binder and ALFA-tag nanobody reference signal (magenta) revealed underlying labeling efficiency. Zoom-ins depict super-resolution representation of target EGFR binders colocalizing with respective ALFA-tag nanobody. (**e**) Quantitative analysis of first nearest neighbor distances (NND) of target-reference signal and fitting the NND data to a two-population model reveals underlying labeling efficiency of target EGFR binders. Data is shown as a box plot, where the median is indicated by the center black line. Boxes extend from the 25th to the 75th percentile of each group's distribution of values. The whiskers extend to points that lie within 1.5 interquartile range of the lower and upper quartile. Data is of n = 6 (GFP-1H1 & GFP-1B2), n = 7 (affibody), n = 11 (EGFR-199.12) and n = 12 (EGFR-D38B1) cells over 3 independent experiments.

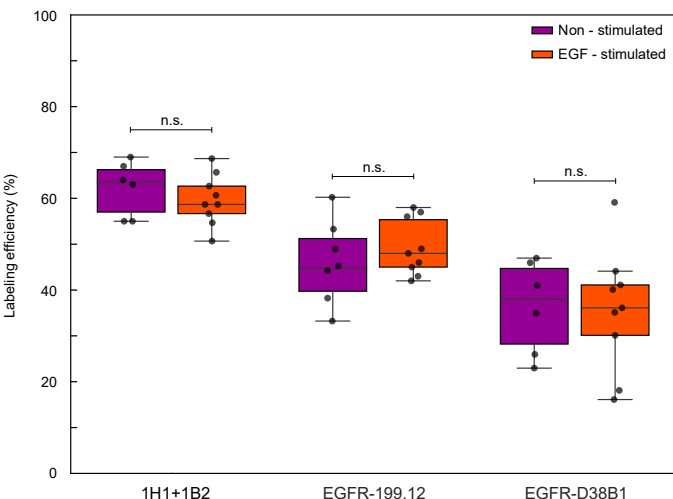

**Extended Data Fig. 8 | Labeling efficiency of EGFR binders is similar for unstimulated and EGF-stimulated EGFRs.** Labeling efficiency for EGFR binders (GFP nanobodies (clone 1H1, clone 1B2), extracellular anti-EGFR antibody + 2[ary] nanobody, intracellular anti-EGFR antibody + 2[ary] nanobody) were determined simultaneously in unstimulated and EGF-stimulated CHO cells expressing EGFR-ALFA-mEGFP on the cell surface. Quantitative analysis of first nearest neighbor distances (NND) of target-reference signal and fitting the NND data to a two-population model reveals underlying labeling efficiency of target EGFR binders. Data is shown as a box plot, where the median is indicated by the center black line. Boxes extend from the 25th to the 75th percentile of each group's distribution of values. The whiskers extend to points that lie within 1.5 interquartile range of the lower and upper quartile. Data is of n = 9 (EGF-stimulated) and n = 6 (non-stimulated) cells over 3 independent experiments and was tested for statistical significance via a two-sided bootstrap ratio test. n.s., non-significant.

# Reporting Summary

## Statistics

For all statistical analyses, confirm that the following items are present in the figure legend, table legend, main text, or Methods section.

| n/a | Confirmed | |
|---|---|---|
| ☐ | ☒ | The exact sample size (*n*) for each experimental group/condition, given as a discrete number and unit of measurement |
| ☐ | ☒ | A statement on whether measurements were taken from distinct samples or whether the same sample was measured repeatedly |
| ☐ | ☒ | The statistical test(s) used AND whether they are one- or two-sided |
| | | *Only common tests should be described solely by name; describe more complex techniques in the Methods section.* |
| ☒ | ☐ | A description of all covariates tested |
| ☒ | ☐ | A description of any assumptions or corrections, such as tests of normality and adjustment for multiple comparisons |
| ☐ | ☒ | A full description of the statistical parameters including central tendency (e.g. means) or other basic estimates (e.g. regression coefficient) AND variation (e.g. standard deviation) or associated estimates of uncertainty (e.g. confidence intervals) |
| ☐ | ☒ | For null hypothesis testing, the test statistic (e.g. *F*, *t*, *r*) with confidence intervals, effect sizes, degrees of freedom and *P* value noted |
| | | *Give P values as exact values whenever suitable.* |
| ☒ | ☐ | For Bayesian analysis, information on the choice of priors and Markov chain Monte Carlo settings |
| ☒ | ☐ | For hierarchical and complex designs, identification of the appropriate level for tests and full reporting of outcomes |
| ☒ | ☐ | Estimates of effect sizes (e.g. Cohen's *d*, Pearson's *r*), indicating how they were calculated |

*Our web collection on statistics for biologists contains articles on many of the points above.*

## Software and code

Policy information about availability of computer code

| Data collection | Raw microscopy data was acquired using µManager (Version 2.0-gamma) (Edelstein, A., Amodaj, N., Hoover, K., Vale, R. & Stuurman, N. Curr. Protoc. Mol. Biol. 14.20 (2010)) |
|---|---|
| Data analysis | All data were analyzed using the open source software Picasso (Versions 0.6.0 - 0.6.5) (Schnitzbauer, J., Strauss, M. T., Schlichthaerle, T., Schueder, F., & Jungmann, R. (2017). Super-resolution microscopy with DNA-PAINT. Nature Protocols, 12(6), 1198–1228. http://doi.org/10.1038/nprot.2017.024). Latest Version can be found on GitHub: https://github.com/jungmannlab/picasso |

For manuscripts utilizing custom algorithms or software that are central to the research but not yet described in published literature, software must be made available to editors and reviewers. We strongly encourage code deposition in a community repository (e.g. GitHub). See the Nature Portfolio guidelines for submitting code & software for further information.

## Data

Policy information about availability of data

All manuscripts must include a data availability statement. This statement should provide the following information, where applicable:
- Accession codes, unique identifiers, or web links for publicly available datasets
- A description of any restrictions on data availability
- For clinical datasets or third party data, please ensure that the statement adheres to our policy

All raw data is available upon reasonable request from the authors.

# Human research participants

Policy information about studies involving human research participants and Sex and Gender in Research.

| Reporting on sex and gender | n/a |
|---|---|
| Population characteristics | n/a |
| Recruitment | n/a |
| Ethics oversight | n/a |

Note that full information on the approval of the study protocol must also be provided in the manuscript.

# Field-specific reporting

Please select the one below that is the best fit for your research. If you are not sure, read the appropriate sections before making your selection.

☒ Life sciences    ☐ Behavioural & social sciences    ☐ Ecological, evolutionary & environmental sciences

For a reference copy of the document with all sections, see nature.com/documents/nr-reporting-summary-flat.pdf

# Life sciences study design

All studies must disclose on these points even when the disclosure is negative.

| Sample size | Sample size n is defined as the number of cells that were recorded and analyzed. |
|---|---|
| Data exclusions | No data were excluded. |
| Replication | All replications were successful. |
| Randomization | N/A, no grouping of experiments or samples were performed. |
| Blinding | N/A, no grouping of experiments or samples were performed. |

# Reporting for specific materials, systems and methods

We require information from authors about some types of materials, experimental systems and methods used in many studies. Here, indicate whether each material, system or method listed is relevant to your study. If you are not sure if a list item applies to your research, read the appropriate section before selecting a response.

## Materials & experimental systems

| n/a | Involved in the study |
|---|---|
| ☐ | ☒ Antibodies |
| ☐ | ☒ Eukaryotic cell lines |
| ☒ | ☐ Palaeontology and archaeology |
| ☒ | ☐ Animals and other organisms |
| ☒ | ☐ Clinical data |
| ☒ | ☐ Dual use research of concern |

## Methods

| n/a | Involved in the study |
|---|---|
| ☒ | ☐ ChIP-seq |
| ☒ | ☐ Flow cytometry |
| ☒ | ☐ MRI-based neuroimaging |

# Antibodies

| Antibodies used | 1) Anti-PD-L1 Rat Monoclonal, BioLegend (cat: 124302, clone 10F.9G2), dilution 50 nM<br>2) Anti-CD80 Armenian Hamster Monoclonal, BioLegend (cat: 104702, clone 16-10A1), dilution 50 nM<br>3) Anti-CD86 Rat Monoclonal, BioLegend (cat: 105002, clone GL-1 ), dilution 50 nM<br>4) Anti-EGFR Mouse Monoclonal, Thermo Fisher Scientific (cat: MA5-13319, clone 199.12), dilution 1 in 100<br>5) Anti-EGFR  Monoclonal, Affibody (Abcam, clone ab81872, unconjugated), dilution 1 in 100<br>6) Anti-ALFA Nanobody, NanoTag (cat: N1505, clone 1G5 ), dilution 0.5 - 50 nM<br>7) Anti-GFP Nanobody, NanoTag (cat: N0305, clone: 1H1), dilution 50 nM<br>8) Anti-Mouse Ig kappa light chain Nanobody, NanoTag (cat: N1205, clone 1A23), dilution 50 nM |
|---|---|

9) Anti-Rabbit IgG Nanobody, NanoTag (cat: N2405, clone 10E10), dilution 50 nM
10) Anti-RFP Nanobody, NanoTag (cat: N0405, clone 2B12), dilution 50 nM
11) Anti-TagFP Nanobody, NanoTag (cat: N0505, clone 1H7), dilution 50 nM
12) Anti-mEos Nanobody, NanoTag (cat: N3105, clone 1E8), dilution 50 nM
13) Anti-mNeonGreen Nanobody, NanoTag (cat: N3205, clone 1E2), dilution 50 nM
14) Anti-SPOT Nanobody, ChromoTek (cat: etb, RRID_2827572), dilution 50 nM
15) Anti-EGFR, monoclonal rabbit IgG, Cell Signaling, (cat: 4267, clone: D38B1), dilution 1 in 100
16) Anti-GFP Nanobody, NanoTag, (custom, clone 1B2, C-terminal cysteine), dilution 50 nM

| Validation | All antibodies and nanobodies were validated by the manufacturers using western blot and immunofluorescence. |

# Eukaryotic cell lines

Policy information about cell lines and Sex and Gender in Research

| Cell line source(s) | CHO-K1 (Sigma Aldrich, 85051005-1VL), BSC1 (Creative-Bioarray, CSC-C9342L), U-2 OS-CRISPR-Nup96-mEGFP cells were obtained from the Ellenberg and Ries lab (Reference: https://doi.org/10.1038/s41592-019-0574-9). |

| Authentication | The cell lines were not authenticated. |

| Mycoplasma contamination | All cell lines have been tested negative for mycoplasma contamination. |

| Commonly misidentified lines (See ICLAC register) | No commonly misidentified cell lines were used. |

