## [Peer Review File · Nature Methods]

Peer Review Information

Manuscript Title: Quantification of absolute labeling efficiency at the single-protein level

Corresponding author name(s): Ralf Jungmann

Editorial Notes: None

Reviewer Comments & Decisions:

Decision Letter, initial version:

Dear Ralf,

Thank you for your letter detailing how you would respond to the reviewer concerns regarding your Brief Communication, "Quantification of absolute labeling efficiency at the single-protein level". We have decided to invite you to revise your manuscript as you have outlined, before we reach a final decision on publication.

[Redacted]

Note: This URL links to your confidential home page and associated information about manuscripts

you may have submitted, or that you are reviewing for us. If you wish to forward this email to co-authors, please delete the link to your homepage.

We hope to receive your revised paper within three months. If you cannot send it within this time, please let us know. In this event, we will still be happy to reconsider your paper at a later date so long as nothing similar has been accepted for publication at Nature Methods or published elsewhere.

OPEN SCIENCE REQUIREMENTS

REPORTING SUMMARY AND EDITORIAL POLICY CHECKLISTS

IMAGE INTEGRITY

When submitting the revised version of your manuscript, please pay close attention to our Digital Image Integrity Guidelines and to the following points below:

DATA AVAILABILITY

CODE AVAILABILITY

Please include a "Code Availability" subsection in the Online Methods which details how your custom code is made available. Only in rare cases (where code is not central to the main conclusions of the paper) is the statement "available upon request" allowed (and reasons should be specified).

For more information on our code sharing policy and requirements, please see: <https://www.nature.com/nature-research/editorial-policies/reporting-standards#availability-of-computer-code>

MATERIALS AVAILABILITY

****SUPPLEMENTARY PROTOCOL**** Please include a step-by-step protocol for a "standard" labeling scheme.

To help facilitate reproducibility and uptake of your method, we ask you to prepare a step-by-step Supplementary Protocol for the method described in this paper. We encourage authors to share their step-by-step experimental protocols on a protocol sharing platform of their choice and report the protocol DOI in the reference list. Nature Portfolio's Protocol Exchange is a free-to-use and open resource for protocols; protocols deposited in Protocol Exchange are citable and can be linked from the published article. More details can found at www.nature.com/protocolexchange/about.

ORCID

Nature Methods is committed to improving transparency in authorship. As part of our efforts in this direction, we are now requesting that all authors identified as 'corresponding author' on published papers create and link their Open Researcher and Contributor Identifier (ORCID) with their account on the Manuscript Tracking System (MTS), prior to acceptance. This applies to primary research papers only. ORCID helps the scientific community achieve unambiguous attribution of all scholarly contributions. You can create and link your ORCID from the home page of the MTS by clicking on 'Modify my Springer Nature account'. For more information please visit please visit www.springernature.com/orcid.

Sincerely,
Rita

Rita Strack, Ph.D.
Senior Editor
Nature Methods

Reviewers' Comments:

Reviewer #1:

Remarks to the Author:

Hellmeier et al. introduce a method that provides absolute labeling efficiency for a specific membrane protein using a reference tag fused to the target protein. Applying exchange DNA-PAINT with two different docking and imager strands the authors correlated the presence of reference tag and target tag binders to calculate the labeling efficiency. In a first set of experiments the authors transiently infected cells with CD86 carrying an Alfa-tag as reference at the N-terminus (extracellular) and used GFP as target tag at the C-terminus (intracellular). They used an anti-Alfa-tag nanobody and an anti mEGFP nanobody labeled with a docking strand and performed exchange DNA-PAINT to determine the number of bound nanobodies determining the cross-nearest neighbor distance (NND) of each reference signal to its nearest target binder. Interestingly, they observed significantly different labeling efficiencies of target nanobodies, ranging from almost 50 % for anti-GFP (clone 1H1) to below 10 % for anti-mEOS2 (clone 1E8). Next, they investigated the labeling efficiency of primary antibodies directed against different membrane proteins using a quite complex approach, i.e. they used two enzymatic site-specific conjugation approaches targeting the Fc region of the primary antibodies. Intracellular they fused an Alfa-tag as reference tag and a mEGFP (just as transfection marker). Then they labeled the two enzymatically generated labeling positions with two docking strand tagged nanobodies and the Alfa-tag with one docking strand tagged nanobody for quantification. Finally, they aimed to demonstrate the importance of absolute quantification of labeling efficiencies for the

investigation of the oligomerization of the epidermal growth factor receptor (EGFR), which dimerizes upon EGF stimulation. Therefore, they expressed EGFR-Alfa-mEGFP (Alfa: extracellular; mEGFP intracellular) and investigated the labeling efficiency of different EGFR binders (antibodies, affibody and GFP nanobodies 1H1 and 1B2). However, it is also known that EGFR can form higher order multimers upon activation, whose nature and function are poorly understood. The presence of higher order multimers was completely ignored in data analysis, which is certainly a problem of the present manuscript.

Overall, the manuscript presents a new method that can determine the labeling efficiency for a selected protein under specific labeling and fixation conditions. The presented data are interesting and solid but the question arises what the data are good for? For example, the target tag can exhibit completely different accessibility fused to other proteins or located either intra- or extracellular to the same protein, and the fixation conditions might impact the labeling efficiency as well. Hence, a general applicability is not given and the advantage of the presented method remains limited. The authors themselves state that the labeling efficiency is different for GFP fused to two different membrane proteins. Furthermore, maturation of FPs might differ for different fusion proteins and cell culture conditions furthermore complicating data interpretation.

In addition, the method involves "overlabeling" of proteins with tags and several binders (usually 3-4 per membrane protein) and it remains to be tested how such complex modifications impact protein maturation and natural behavior in the membrane, e.g. dimerization or multimerization. Even the transfection efficiency might influence labeling accessibility and multimerization. Such external factors will, however, be very difficult to consider in data analysis. Ultimately, I do not see the benefit of the proposed method for quantitative super-resolution microscopy.

Finally, I disagree with the authors that we are currently lacking methods to quantify labeling efficiency. DNA-origamis have drawbacks as mentioned by the authors but considering the apparent limitations listed above I believe they can still be used as quantification references. See, for example: Zanicchi, F., Manzo, C., Alvarez, A. et al. A DNA origami platform for quantifying protein copy number in super-resolution. *Nat Methods* 14, 789–792 (2017). <https://doi.org/10.1038/nmeth.4342>

Or, a protein based quantification reference:

Kieran Finan, Anika Raulf, Mike Heilemann. A Set of Homo-Oligomeric Standards Allows Accurate Protein Counting. *Angew. Chem. Int. Ed.* 54, 12049-1205 (2015). <https://doi.org/10.1002/anie.201505664>

Reviewer #2:

Remarks to the Author:

Review of "Quantification of absolute labeling efficiency at the single-protein level" by Hellmeier et al.

Introduction

In their manuscript, Hellmeier et al. introduce a simple and effective method for measuring the labeling efficiency of antibodies, nanobodies, and other tags used in fluorescence microscopy. The new concept, based on tagging a protein with two tags, one of which acts as a reference marker, allows a direct measurement of the absolute fraction of a given target tag which is labeled with a fluorophore in the prepared sample. This approach may be used for comparing the efficiency of different tags, as the authors demonstrate e.g. by quantitatively comparing a number of nanobodies directed against various target tags (GFP, RFP, Alfa-tag, etc). The authors further evaluated the labeling efficiency of

DNA-conjugated antibodies. Finally, the great utility of quantifiable labeling efficiencies was demonstrated by measuring EGFR oligomerization in CHO cells.

General comments

The method presented in this study is simple and elegant, and I agree with the authors' view that quantifiable labeling efficiency is a highly relevant topic, especially given current developments in super-resolution fluorescence microscopy. The beauty of the concept lies in the recognition that the labeling processes of the reference tag and the target tag are independent, and hence during the analysis one can simply discard all proteins which were not tagged at the reference site. I do see some possible caveats to the method, however, and I would suggest for these questions to be addressed by the authors in the paper. Please see my specific comments, below.

Specific comments

1. The analysis of the labeling efficiency fundamentally depends on identifying how many reference markers are associated with a target marker. The authors did not use a simple counting approach, however, and instead simulated possible distributions of cross-nearest-neighbor distances (NND), and solved for the labeling efficiency based on these simulations.

Fundamentally, the counting problem would seem to become more difficult when the density of reference markers becomes higher. In this case, it is no longer possible to clearly discard the population of proteins which are tagged only on the target site. This will be true regardless of the analysis method used.

For their work, the authors used proteins which were expressed on the surface of a cell, at a density for which the reference markers were well-separated. The results from these experiments is clear and convincing, and I think this represents a reliable measure of labeling efficiency for the tags tested on a cell surface.

Given the statements above, what would happen when the density of the protein of interest is higher? For example, could this technique be used to quantify the labeling efficiency of an antibody against an arbitrary protein within the cell? Going further, could the labeling efficiency be measured in the context of that protein, (e.g. within the nuclear pore complex) rather than at the cell surface?

This question is essentially twofold: (a) How does reference / target tag density affect the measurement, and (b) can the labeling efficiency be measured within the cellular context, rather than at a membrane surface.

2. The authors used a simulation of the distribution of NNDs to determine the labeling efficiency of the target. However, the results of these simulations would seemingly depend on the spatial density of the reference tag. How confident are the authors that the simulations used in their analysis accurately reflect the real situation? Again, this problem would seem to become more difficult when the density of the reference is higher.

3. The analysis concept is based on the assumption that the labeling of the target tag is independent of the labeling of the reference tag. Is this necessarily true? In other words, could the labeling of the reference site cause a change in the probability of labeling the target site, or vice versa? Is there a

way to control for this effect?

4. Can the authors validate the quantitative measurements of labeling efficiency against another, established approach? For example, using one of the methods mentioned in references 12-14?

5. The authors did not give a name to their method. Despite the ubiquity of acronyms in the super-resolution fluorescence field, I would recommend finding a suitable name for the new approach.

Conclusion

The manuscript presents a compelling new method, which the authors have applied to measure the absolute labeling efficiency of nanobody tags (and others) which are commonly used in super-resolution fluorescence microscopy. The method and results could easily constitute two high-profile publications. However, the new approach does have possible caveats, as discussed above, particularly concerning limitations at high reference marker density. I recommend publication in Nature Methods once the reviewer comments have been addressed.

Reviewer #3:

Remarks to the Author:

This manuscript by Hellmeier et al. presents a new way to quantify the absolute binder (antibodies, nanobodies, etc.) labeling efficiency. To realize this, The authors first introduced a reference tag to the targets. Then, both the reference and target were imaged by DNA PAINT technology. The absolute labeling efficiency was determined by simply calculating the percentage of reference labels that were colocalized with target labels. Compared to the other methods using structures of well-defined geometry (nuclear pore proteins or DNA origami), the proposed method is much easier to be applied by just introducing a reference tag to the target and 2 color super resolution imaging is performed to image both the target and reference labels. A number of points must be addressed and some details added before being able to recommend acceptance for publication.

Major Comments:

1. The labeling efficiency determination equation ignores many experimental conditions. For example, how about the effect of nonspecific binding of reference labeling? If tags were introduced to both targets and references to determine the absolute labeling efficiency of the targets, I assume that the tagging efficiency for target and reference should be the same to be able to determine the absolute labeling efficiency of the targets.

2. Will the labeling efficiency of the targets be affected by additionally introducing the reference labels? It is better to cross validate the current methods with other methods to determine the labeling efficiency, e.g. using the structure with the well-defined geometry such DNA origami and nuclear pore protein.

3. The presented method was only evaluated by using DNA PAINT. Further, only the labeling efficiency of the primary antibody was demonstrated. Considering that primary antibody is normally quite expensive and DNA PAINT labeling and imaging is still not easy for most labs (commercially available DNA PAINT kits are limited to certain antibodies), can the author comment or even demonstrate on the potential of the conventional dSTORM imaging and secondary antibody labeling to be used in the presented method?

4. The author only demonstrated in the dimerization of the same protein EGFR using the presented

method, it would be great if the author could quantitatively compare the stoichiometry of different proteins using the current method.

Minor Comments:

1. The simulation for the labeling efficiency needs the density of target and reference monomers. However, in the real experiments, the density can be different for different cells and even in different areas of a single cell. Can the authors comment on this? The automatic determination of labeling efficiency using cross-nearest neighbor distance (NND) is still not clear for me. Can the author show NND histograms curves of different labeling efficiencies used to fit the experimental distribution? Since such curves are highly dependent on the density parameters, I think it is not applicable for cells with different target and reference density.
2. The author showed that there was a stark difference between transglutaminase and GlyCLICK in the case of the anti-CD80 antibody without any further explanation.

Author Rebuttal to Initial comments

Reviewers' Comments:

Reviewer #1:

Remarks to the Author:

Hellmeier et al. introduce a method that provides absolute labeling efficiency for a specific membrane protein using a reference tag fused to the target protein. Applying exchange DNA-PAINT with two different docking and imager strands the authors correlated the presence of reference tag and target tag binders to calculate the labeling efficiency. In a first set of experiments the authors transiently infected cells with CD86 carrying an Alfa-tag as reference at the N-terminus (extracellular) and used GFP as target tag at the C-terminus (intracellular). They used an anti-Alfa-tag nanobody and an anti mEGFP nanobody labeled with a docking strand and performed exchange DNA-PAINT to determine the number of bound nanobodies determining the cross-nearest neighbor distance (NND) of each reference signal to its nearest target binder. Interestingly, they observed significantly different labeling efficiencies of target nanobodies, ranging from almost 50 % for anti-GFP (clone 1H1) to below 10 % for anti-mEOS2 (clone 1E8).

Next, they investigated the labeling efficiency of primary antibodies directed against different membrane proteins using a quite complex approach, i.e. they used two enzymatic site-specific conjugation approaches targeting the Fc region of the primary antibodies. Intracellular they fused an Alfa-tag as reference tag and a mEGFP (just as transfection marker). Then they labeled the two enzymatically generated labeling positions with two docking strand tagged nanobodies and the Alfa-tag with one docking strand tagged nanobody for quantification. Finally, they aimed to demonstrate the importance of absolute quantification of labeling efficiencies for the investigation of the oligomerization of the epidermal growth factor receptor (EGFR), which dimerizes upon EGF stimulation. Therefore, they expressed EGFR-Alfa-mEGFP (Alfa: extracellular; mEGFP intracellular) and investigated the labeling efficiency of different EGFR binders (antibodies, affibody and GFP nanobodies 1H1 and 1B2). However, it

is also known that EGFR can form higher order multimers upon activation, whose nature and function are poorly understood. The presence of higher order multimers was completely ignored in data analysis, which is certainly a problem of the present manuscript.

Overall, the manuscript presents a new method that can determine the labeling efficiency for a selected protein under specific labeling and fixation conditions. The presented data are interesting and solid but the question arises what the data are good for? For example, the target tag can exhibit completely different accessibility fused to other proteins or located either intra- or extracellular to the same protein, and the fixation conditions might impact the labeling efficiency as well. Hence, a general applicability is not given and the advantage of the presented method remains limited. The authors themselves state that the labeling efficiency is different for GFP fused to two different membrane proteins. Furthermore, maturation of FPs might differ for different fusion proteins and cell culture conditions furthermore complicating data interpretation.

In addition, the method involves “overlabeling” of proteins with tags and several binders (usually 3-4 per membrane protein) and it remains to be tested how such complex modifications impact protein maturation and natural behavior in the membrane, e.g. dimerization or multimerization. Even the transfection efficiency might influence labeling accessibility and multimerization. Such external factors will, however, be very difficult to consider in data analysis. Ultimately, I do not see the benefit of the proposed method for quantitative super-resolution microscopy.

Finally, I disagree with the authors that we are currently lacking methods to quantify labeling efficiency. DNA-origamis have drawbacks as mentioned by the authors but considering the apparent limitations listed above I believe they can still be used as quantification references. See, for example:

Zanacchi, F., Manzo, C., Alvarez, A. et al. A DNA origami platform for quantifying protein copy number in super-resolution. *Nat Methods* 14, 789–792 (2017). <https://doi.org/10.1038/nmeth.4342>

Or, a protein based quantification reference:

Kieran Finan, Anika Raulf, Mike Heilemann. A Set of Homo-Oligomeric Standards Allows Accurate Protein Counting. *Angew. Chem. Int. Ed.* 54, 12049-1205 (2015). <https://doi.org/10.1002/anie.201505664>

We thank the reviewer for his/her feedback and aim to clarify and address several aspects of our study, which have potentially led to some misunderstandings.

We opted to first assess our approach on widely employed tag-proteins, utilizing a cargo protein that is monomeric and randomly distributed in transiently transfected cells (e.g. CHO cells). We emphasize that our choice of using well-characterized tag-proteins as well as membrane proteins for evaluation was guided by their widespread use and attention within multiple scientific communities. The intention was to provide a practical and straightforward method for quantification in an unbiased and automated way (see also Supplementary Figure 1).

Importantly, as depicted in our Supplementary Figure 4, we were indeed mindful of the potential impact of intra- vs. extracellular positioning of the tags on our quantified labeling efficiency (LE). In response to this reviewer's comment, we repositioned the reference and target as well as conducted our analysis not only on a single membrane protein but included in addition to CD86 another randomly distributed membrane protein, CD70. We find that the resulting labeling efficiency remains consistent regardless of the cargo protein utilized as well as tag protein positioning, whether placed on the intracellular or extracellular side. Specifically, the case of CD86 vs. CD70 yielded identical labeling efficiency values for GFP nanobody (clone 1H1) across the four distinct scenarios, further validating the robustness of our approach. Detailed results for the four construct variations, with mEGFP oriented towards either the intracellular or extracellular side of the membrane, can now be found in Supplementary Table 1.

While the maturation of fluorescent proteins can vary based on fusion proteins and conditions, we disagree that these intricacies might invalidate our labeling efficiency evaluation method, as these represent real-world scenarios where experiments are conducted. However, it is crucial to emphasize the significance of careful consideration when designing plasmids to ensure sufficient spacing between the reference and target sites or, alternatively, by adding additional spacers. These precautions are essential to ensure both target and reference binders can bind without any restrictions.

Furthermore, we acknowledge the challenges inherent to site-specific conjugation approaches for primary antibodies (Figure 2), which might not be straightforward and applicable in all laboratories. As a commonly preferred alternative, we use a method based on secondary labeling of primary antibodies using 2^{ary} nanobodies. However, it's important to note that we applied direct DNA-conjugation to the Fc portion of primary antibodies in Figure 2. In contrast, for Figure 3, we opted for secondary labeling of primary antibodies via 2^{ary} nanobodies, each tethered to a single DNA-PAINT docking strand.

The comment on EGFR dimerization is well-taken. We decided on this prominent member of the RTK family due to EGFR's well-studied nature and its confirmed dimerization upon EGF ligand addition¹⁻⁵. However, as noted by the reviewer, there is to date still no consensus yet regarding the exact oligomeric state of EGFR although major effort has been taken in the last decade to unravel this conundrum. So far, varying results have been observed in different studies, likely due to differences in EGFR stimulation⁶⁻¹¹, cell systems^{6-8,10,11}, and the distinction between live-cell and fixed-cell conditions^{6,7,9-11}, both in the inactive and active states of EGFR. The diverse outcomes highlight the complexity of EGFR oligomerization and the need for further biological investigation.

*However, we do note that this goes beyond the scope of our current study and could be tackled in a follow-up investigation. However, we conducted a detailed analysis of our data to explore the presence of higher-order oligomers using N-th nearest neighbor distance (NND) analysis as illustrated in **Figure R1**. Our findings revealed no indications for higher order oligomers than dimers as can be seen in the figure below. This observation could be attributed to our focus on utilizing cell areas for analysis, particularly with an EGFR density below 400 molecules per μm^2 . However, we believe that the focus on the amount of dimerization is a valid assumption in our case and does not lead to issues in the interpretation on our*

results, which does not focus on the biological implication of EGFR oligomerization, but rather the fact that labeling efficiency corrected experiments with different labeling molecules lead to similar amounts of dimerization in treated and untreated cells.

Figure R1 | EGFR dimerization upon EGF addition. 3 different binders (**a**, anti-GFP 1H1 + 1B2 nanobody (cyan), **b**, extracellular anti-EGFR antibody + anti-mouse kappa light chain nanobody (yellow), **c**, intracellular anti-EGFR antibody + anti-rabbit IgG nanobody (red)) with different labeling efficiencies were used for labeling transiently transfected CHO-EGFR-ALFA-mEGFP cells. Schematics of the construct for EGFR dimers is shown in the insets. While first nearest neighbor distances (1st NNDs) of EGFR (histograms of the distances and kernel density estimation are displayed) indicate a significant population of EGFR dimers at a distinct distance (d_{dimer}), 2nd to 3rd NND distributions are distributed according to Complete Spatial Randomness (CSR) for all three binders at a density-based distance (d_{CSR}), thus providing no evidence of higher order protein complexes. CSR simulations (solid lines) were conducted at experimentally determined protein densities. Data is from ≥ 2 independent experiments.

With regards to the utility of our approach, we acknowledge the existence of established methods that use either DNA origami or the nuclear pore complex for assessing binder quality. However, it is important to highlight that our approach focuses on determining labeling LE under identical cell culture conditions as the actual experiment, thereby enabling binder characterization within the native cellular context of a broader range of potential targets as well as references. Although we used for the current study membrane proteins as cargo proteins, we are not restricted to the plasma membrane as long as we achieve single protein resolution. For this purpose, we included an intracellular target like Nup96-mEGFP to evaluate the labeling efficiency of GFP nanobody (clone 1H1). Here, we exploited the well-characterized nuclear pore complex as a reference structure for direct counting of labeled Nup96-mEGFP subunits yielding similar LE values, as illustrated in Supplementary Fig. 3. This comparison demonstrates the robustness and reliability of our method.

Lastly, we appreciate the note on potential concerns in Figure 3 regarding the targeting of multiple epitopes and locations on a single protein. To this end, we confirmed that individual labeling efficiency

values were comparable for single and multiple binder incubations, as well as under non-stimulated and stimulated conditions, as highlighted in Supplementary Table 3 and Supplementary Table 4. However, for all binder-targeted proteins exclusively monoclonal primary antibodies or monovalent nanobodies were used inhibiting 'overlabeling' of target proteins and ensuring 1:1 ratio of target to reference.

Reviewer #2:

Remarks to the Author:

Review of “Quantification of absolute labeling efficiency at the single-protein level” by Hellmeier et al.

Introduction

In their manuscript, Hellmeier et al. introduce a simple and effective method for measuring the labeling efficiency of antibodies, nanobodies, and other tags used in fluorescence microscopy. The new concept, based on tagging a protein with two tags, one of which acts as a reference marker, allows a direct measurement of the absolute fraction of a given target tag which is labeled with a fluorophore in the prepared sample. This approach may be used for comparing the efficiency of different tags, as the authors demonstrate e.g. by quantitatively comparing a number of nanobodies directed against various target tags (GFP, RFP, Alfa-tag, etc). The authors further evaluated the labeling efficiency of DNA-conjugated antibodies. Finally, the great utility of quantifiable labeling efficiencies was demonstrated by measuring EGFR oligomerization in CHO cells.

General comments

The method presented in this study is simple and elegant, and I agree with the authors' view that quantifiable labeling efficiency is a highly relevant topic, especially given current developments in super-resolution fluorescence microscopy. The beauty of the concept lies in the recognition that the labeling processes of the reference tag and the target tag are independent, and hence during the analysis one can simply discard all proteins which were not tagged at the reference site. I do see some possible caveats to the method, however, and I would suggest for these questions to be addressed by the authors in the paper. Please see my specific comments, below.

We are grateful for the very supportive and thorough review of our work by this reviewer.

Specific comments

1. The analysis of the labeling efficiency fundamentally depends on identifying how many reference markers are associated with a target marker. The authors did not use a simple counting approach, however, and instead simulated possible distributions of cross-nearest-neighbor distances (NND), and solved for the labeling efficiency based on these simulations. Fundamentally, the counting problem would seem to become more difficult when the density of reference markers becomes higher. In this case, it is no longer possible to clearly discard the population of proteins which are tagged only on the target site.

This will be true regardless of the analysis method used. For their work, the authors used proteins which were expressed on the surface of a cell, at a density for which the reference markers were well-separated. The results from these experiments are clear and convincing, and I think this represents a reliable measure of labeling efficiency for the tags tested on a cell surface. Given the statements above, what would happen when the density of the protein of interest is higher? For example, could this technique be used to quantify the labeling efficiency of an antibody against an arbitrary protein within the cell? Going further, could the labeling efficiency be measured in the context of that protein, (e.g. within the nuclear pore complex) rather than at the cell surface?

This question is essentially twofold: (a) How does reference / target tag density affect the measurement, and (b) can the labeling efficiency be measured within the cellular context, rather than at a membrane surface.

We agree with the reviewer that understanding the impact of reference/target tag density on our method is crucial and are grateful to the reviewer for pointing us into this direction. Our approach – compared to straightforward counting of colocalizing reference-target signals at a given offset distance d_{off} (as illustrated in Figure 1) – indeed considers simulated cross-nearest-neighbor distances, thus circumventing the challenges associated with density fluctuations. The concept is that this approach can provide an accurate estimation of labeling efficiency even in scenarios where counting-based techniques falter due to increased density. In the case of “simple counting”, as density increases, the contributions from “monomers” and “hetero-dimers” start to overlap progressively, resulting in an overestimation of the underlying labeling efficiency. Consequently, the accuracy of the labeling efficiency estimation using simple counting diminishes with rising density. To quantify this distinction, we now include an additional supplementary figure (see Supplementary Figure 1 for further details) depicting the robustness of labeling efficiency evaluation for our method compared to the labeling efficiency values determined via simple counting at a given cutoff distance as densities shift from low to high. This underscores our approach's capability to ensure reliable and unbiased labeling efficiency determinations regardless of density fluctuations.

As for the feasibility of measuring labeling efficiency within different cellular compartments, we acknowledge the importance of clarifying this aspect. In theory, there are no intrinsic limitations to our approach, provided that molecular resolution can be achieved and the target protein is monomeric and mainly randomly distributed. However, to substantiate the feasibility of measuring labeling efficiency within various cellular compartments, we recognize the need to provide experimental evidence. To this end, we performed an additional experiment focusing on the nuclear pore complex (NPC), specifically Nup96 as protein target.

The experimental workflow is as follows:

- 1. The nuclear pore complex (NPC) protein Nup96 is tagged with mEGFP. Subsequently, Nup96 is site-specifically labeled using GFP nanobody (clone 1H1).*
- 2. Following the labeling procedure, we performed high-resolution DNA-PAINT microscopy, which allowed us to resolve single Nup96 copies, subsequently enabling us to directly count the number of labeled protein copies in the NPC. This allowed us to compare the obtained labeling efficiency by our membrane assay to the direct counting of proteins in the NPC. Furthermore, we compared our derived labeling efficiency for GFP-1H1 for NUP96-mEGFP to those determined for CD86-mEGFP and CD70-mEGFP. In both cases, we achieved results, which are in very good agreement with each other*
- 3. Finally, we compared the labeling efficiency values obtained through determining the number of GFP-1H1 signals per NUP96 ring (2 rings, 8-fold symmetry with 2 possible GFP-1H1 per unit) to those derived using a previously established method from Jonas Ries' lab (<https://www.nature.com/articles/s41592-019-0574-9>). This comparative analysis ensures a comprehensive assessment of our method's accuracy and reliability in the context of the nuclear pore complex.*

Through this revised explanation, we aim to accurately convey the advantage of our method in maintaining consistent labeling efficiency outcomes, even in scenarios characterized by varying density levels as well as provide a clearer understanding of the effectiveness of our approach within intricate intracellular environments, such as the nuclear pore complex, and its comparison with existing evaluation techniques.

2. The authors used a simulation of the distribution of NNDs to determine the labeling efficiency of the target. However, the results of these simulations would seemingly depend on the spatial density of the reference tag. How confident are the authors that the simulations used in their analysis accurately reflect the real situation? Again, this problem would seem to become more difficult when the density of the reference is higher.

We thank the reviewer for pointing us to this potentially confusing issue. Our initial submission was unfortunately missing Supplementary Information (e.g. Tables 1–5), which helps to answer this question. The simulations employed in our analysis account for several parameters, which are given in detail in the tables in the supplementary section (Supplementary Table 1-4), to ensure unbiased and accurate representation of our data. Importantly, we consider the experimentally determined protein density of both the target and reference markers, achieve for all molecules molecular resolution (<10nm) and account for linkage error and offset of target and reference binder. Additionally, our simulations

incorporate a range of factors, including the complete spatial randomness (CSR) of the target protein distribution.

We have now included an in silico evaluation of our simulations-based analysis in Supplementary Figure 1. We check the self consistency of the algorithm in simulated datasets with a realistic number of molecules and we compare our simulations-based analysis to a simple cutoff method. As the density of the reference tag increases, the distinction between individual binders becomes increasingly more challenging for the cutoff method, however the simulations-based method proves to be robust even at relatively high densities. In any case, our approach utilizes transiently transfected cells which can specifically be selected for moderate expression levels.

In summary, our simulations are structured to accurately reflect the complexities of experimental conditions, factoring in experimental protein densities, spatial randomness, localization precision, linkage error, and binder offset. By taking these parameters into account, we believe that our simulations provide a reliable basis for estimating labeling efficiency across different scenarios, including those with varying reference tag densities.

3. The analysis concept is based on the assumption that the labeling of the target tag is independent of the labeling of the reference tag. Is this necessarily true? In other words, could the labeling of the reference site cause a change in the probability of labeling the target site, or vice versa? Is there a way to control for this effect?

The reviewer is correct that the underlying assumption for our analysis is that the labeling of the target tag is independent of the labeling of the reference tag. This assumption holds particularly true when the distance between the reference and target sites on the same protein is adequately chosen. In our case, we ensured this separation by placing targets either at the extracellular or intracellular side of the plasma membrane, with the reference located on the opposite side.

Notably, Supplementary Figure 4 illustrates that the specific placement of targets and references is not a bias introducing factor, as supported by the case of CD86 vs. CD70 receptor labeling. By swapping respective tag protein positions, we obtained identical results for labeling efficiency between the four different scenarios further solidifying the validity of our approach. Results for the four different constructs facing mEGFP either at the intracellular or extracellular side of the membrane are given in detail in Supplementary Table 1.

It is important to emphasize that careful consideration is required during plasmid design to either maintain a sufficient spatial separation between reference and target sites (as in our case extra- and intracellular space) or to incorporate additional spacers. These measures are essential to guarantee unhindered binding of both target and reference binders.

4. Can the authors validate the quantitative measurements of labeling efficiency against another, established approach? For example, using one of the methods mentioned in references 12-14?

We carried out a thorough validation of our quantitative measurements of labeling efficiency by comparing them with well-established approaches such as recent approaches from the Ries lab (<https://www.nature.com/articles/s41592-019-0574-9>).

To this end, we evaluated the labeling efficiency of GFP nanobody (clone 1H1) in U2OS-CRISPR-Nup96-mEGFP. Here, we exploited the well-characterized nuclear pore complex as a reference structure for direct counting of labeled Nup96-mEGFP subunits. For both scenarios we derived similar results for GFP-1H1 labeling efficiency, as illustrated in the newly added Supplementary Fig. 3.

5. The authors did not give a name to their method. Despite the ubiquity of acronyms in the super-resolution fluorescence field, I would recommend finding a suitable name for the new approach.

We thank the reviewer for this suggestion, but would rather choose to not introduce an additional name for the method at this point.

Conclusion

The manuscript presents a compelling new method, which the authors have applied to measure the absolute labeling efficiency of nanobody tags (and others) which are commonly used in super-resolution fluorescence microscopy. The method and results could easily constitute two high-profile publications. However, the new approach does have possible caveats, as discussed above, particularly concerning limitations at high reference marker density. I recommend publication in Nature Methods once the reviewer comments have been addressed.

Again, thank you very much for your supportive and constructive review of our work.

Reviewer #3:

Remarks to the Author:

This manuscript by Hellmeier et al. presents a new way to quantify the absolute binder (antibodies, nanobodies, etc.) labeling efficiency. To realize this, The authors first introduced a reference tag to the targets. Then, both the reference and target were imaged by DNA PAINT technology. The absolute labeling efficiency was determined by simply calculating the percentage of reference labels that were colocalized with target labels. Compared to the other methods using structures of well-defined geometry (nuclear pore proteins or DNA origami), the proposed method is much easier to be applied by just introducing a reference tag to the target and 2 color super resolution imaging is performed to image both the target and reference labels. A number of points must be addressed and some details added before being able to recommend acceptance for publication.

Thank you very much for the supportive and comprehensive review of our work.

Major Comments:

1. The labeling efficiency determination equation ignores many experimental conditions. For example, how about the effect of nonspecific binding of reference labeling? If tags were introduced to both targets and references to determine the absolute labeling efficiency of the targets, I assume that the tagging efficiency for target and reference should be the same to be able to determine the absolute labeling efficiency of the targets.

We thank the reviewer for pointing us to this issue. We assessed the extent of nonspecific binding for each binder individually within our study, under saturating conditions. Notably, for the sub-stoichiometrically labeled ALFA-tag nanobody, the determined density of unspecific binding is even lower than that indicated in Supplementary Table 10.

Regarding the tagging efficiency: As both tags (reference and target) are on the plasmid, their expression efficiencies are the same by design. Thus this can be safely assumed.

2. Will the labeling efficiency of the targets be affected by additionally introducing the reference labels? It is better to cross validate the current methods with other methods to determine the labeling efficiency, e.g. using the structure with the well-defined geometry such DNA origami and nuclear pore protein.

We thank the reviewer for mentioning this potential issue. We would like to address this concern in two ways, first highlighting our approach's underlying design principles for tag positions: Our methodology assumes independent labeling of both target and reference binders, which is ensured by a finite distance between the reference and target sites. Specifically, the reference is situated either extracellularly or intracellularly, while the target occupies the opposite location. To enable accurate and unrestricted binding, we either ensure substantial spatial separation between the reference and target sites or introduce additional spacers when designing the plasmids. This precaution guarantees that the introduction of reference labels does not compromise the accuracy of target labeling efficiency determination.

Second, we additionally confirmed our results obtained for the GFP nanobody (clone 1H1) labeling efficiency through various other methods. For instance, we employed the NPC as a well-defined structure and compared these results with the binder evaluation method recently introduced by the Ries lab, as depicted in Supplementary Fig. 3.

3. The presented method was only evaluated by using DNA PAINT. Further, only the labeling efficiency of the primary antibody was demonstrated. Considering that primary antibody is normally quite expensive and DNA PAINT labeling and imaging is still not easy for most labs (commercially available DNA PAINT kits are limited to certain antibodies), can the author comment or even demonstrate on the potential of the conventional dSTORM imaging and secondary antibody labeling to be used in the presented method?

We addressed secondary labeling of antibodies, exemplified in Figure 3, wherein a DNA-tagged secondary nanobody is used for labeling the target primary antibody. This configuration effectively characterizes the introduction of primary antibody labeling, followed by secondary nanobody labeling. It is important to note that the resulting labeling efficiency is determined by the product of the primary and secondary binder's individual labeling efficiencies.

In theory, our approach remains universally applicable and isn't restricted solely to DNA-PAINT. Given sufficient spatial resolution and the availability of two distinct signals (reference and target), our method can be adapted to various other imaging techniques. However, as we are not dSTORM experts, conducting experiments within that context would fall beyond the scope of this current paper.

4. The author only demonstrated in the dimerization of the same protein EGFR using the presented method, it would be great if the author could quantitatively compare the stoichiometry of different proteins using the current method.

We appreciate the suggestion of the reviewer to quantify dimer and oligomer formation of further protein targets, however we think that this would be best addressed in a follow-up study, which is focused more on unraveling a novel biological question.

Minor Comments:

1. The simulation for the labeling efficiency needs the density of target and reference monomers. However, in the real experiments, the density can be different for different cells and even in different areas of a single cell. Can the authors comment on this? The automatic determination of labeling efficiency using cross-nearest neighbor distance (NND) is still not clear for me. Can the author show NND histograms curves of different labeling efficiencies used to fit the experimental distribution? Since such curves are highly dependent on the density parameters, I think it is not applicable for cells with different target and reference density.

We thank the reviewer for pointing us to this potentially confusing issue. The density of receptors is a parameter of the simulation that is extracted from the experimental data (number of single receptors / area). Cells were chosen based on a random and monomeric distribution of target proteins and only cells with mostly homogenous densities within the membrane were imaged. For analysis, multiple different cell areas showing homogeneous protein distribution were picked. Furthermore, variations of density within a single cell could be taken into account in more advanced versions of our simulation by using a density mask.

We now further clarify the cross NND analysis with examples of how the simulation and the fit works, and discuss its robustness and advantages over simple counting. We show, as suggested by the reviewer, different labeling efficiencies used to fit the experimental distribution illustrated in Supplementary Figure 1. Our approach, compared to straightforward counting of colocalizing reference-target signals at a given offset distance d_{off} , considers simulated cross-nearest-neighbor distances, thus circumventing the challenges associated with density fluctuations. The concept is that this approach can provide an accurate estimation of labeling efficiency even in scenarios where counting-based techniques falter due to increased density. In the case of “simple counting”, as density increases, the contributions from “monomers” and “hetero-dimers” start to overlap progressively, resulting in an overestimation of the underlying labeling efficiency. Consequently, the accuracy of the labeling efficiency estimation using simple counting diminishes with rising density.

We accounted for multiple critical parameters in our analysis pipeline, which are given in detail in the tables in the supplementary section (Supplementary Table 1-4), to ensure unbiased and accurate representation of our data. Importantly, we consider the experimentally determined protein density of

both the target and reference markers, achieve for all molecules molecular resolution (<10nm) and account for linkage error and offset of target and reference binder. Additionally, our simulations incorporate a range of factors, including the complete spatial randomness (CSR) of the target protein distribution.

2. The author showed that there was a stark difference between transglutaminase and GlyCLICK in the case of the anti-CD80 antibody without any further explanation.

Here, we can only speculate on why GlyCLICK did not perform as well as transglutaminase. GlyCLICK relies first on the hydrolysis of the FC glycan to the inner most GlcNAc (N-Acetylglucosamine) moiety and second on the addition of the GalNAz (N-azidoacetylgalactosamine) to the inner most GlcNAc. Since the structure of the Armenian hamster antibody and its glycan are unknown, there is a possibility that the glycan may not be the correct substrate for the first hydrolysis step. Another possibility could be steric hindrance from the antibody preventing the first or the second step.

References

1. Cho, J. Mechanistic insights into differential requirement of receptor dimerization for oncogenic activation of mutant EGFR and its clinical perspective. *BMB Rep.* **53**, 133–141 (2020).
2. Hynes, N. E. & MacDonald, G. ErbB receptors and signaling pathways in cancer. *Curr. Opin. Cell Biol.* **21**, 177–184 (2009).
3. Linggi, B. & Carpenter, G. ErbB receptors: new insights on mechanisms and biology. *Trends Cell Biol.* **16**, 649–656 (2006).
4. Lemmon, M. A. & Schlessinger, J. Cell signaling by receptor tyrosine kinases. *Cell* **141**, 1117–1134 (2010).
5. Endres, N. F., Barros, T., Cantor, A. J. & Kuriyan, J. Emerging concepts in the regulation of the EGF receptor and other receptor tyrosine kinases. *Trends Biochem. Sci.* **39**, 437–446 (2014).
6. Zanetti-Domingues, L. C. *et al.* The architecture of EGFR's basal complexes reveals autoinhibition mechanisms in dimers and oligomers. *Nat. Commun.* **9**, 4325 (2018).
7. Needham, S. R. *et al.* EGFR oligomerization organizes kinase-active dimers into competent signalling platforms. *Nat. Commun.* **7**, 13307 (2016).
8. Huang, Y. *et al.* Molecular basis for multimerization in the activation of the epidermal growth factor receptor. *Elife* **5**, (2016).
9. Yasui, M., Hiroshima, M., Kozuka, J., Sako, Y. & Ueda, M. Automated single-molecule imaging in living cells. *Nat. Commun.* **9**, 3061 (2018).

10. Sako, Y., Minoghchi, S. & Yanagida, T. Single-molecule imaging of EGFR signalling on the surface of living cells. *Nat. Cell Biol.* **2**, 168–172 (2000).
11. Needham, S. R. *et al.* Measuring EGFR separations on cells with ~10 nm resolution via fluorophore localization imaging with photobleaching. *PLoS One* **8**, e62331 (2013).

Decision Letter, first revision:

Dear Ralf,

Thank you for submitting your revised manuscript "Quantification of absolute labeling efficiency at the single-protein level" (NMETH-A53087A). It has now been seen by the original referees and their comments are below. The reviewers find that the paper has improved in revision, and therefore we'll be happy in principle to publish it in Nature Methods, pending minor revisions to satisfy the referees' final requests and to comply with our editorial and formatting guidelines.

In response to the reviewers, we ask that you further discuss when the results of the assay should be generally applicable and when it should be repeated for specific experimental conditions. We also ask that you discuss whether the approach could be extended beyond a DNA-PAINT readout. As we discussed, please make sure S. Fig 1 is in the resubmission.

Please note, we have promoted your paper to an Article, so you should have plenty of room for discussion. We ask that you reformat as such, with an intro, results, and discussion sections. We think 3,000 words and 3-figures should be appropriate, but we are prepared to be flexible.

TRANSPARENT PEER REVIEW

Please note: we allow redactions to authors' rebuttal and reviewer comments in the interest of confidentiality. If you are concerned about the release of confidential data, please let us know specifically what information you would like to have removed. Please note that we cannot incorporate redactions for any other reasons. Reviewer names will be published in the peer review files if the reviewer signed the comments to authors, or if reviewers explicitly agree to release their name. For more information, please refer to our FAQ page.

ORCID

Sincerely,
Rita

Rita Strack, Ph.D.
Senior Editor
Nature Methods

Reviewer #1 (Remarks to the Author):

I greatly appreciate the effort made by the authors to resolve my doubts about the general usefulness of the method. As already mentioned in my first assessment I do not question the results presented but I still disbelief that the method will be valued and extensively used by the community. I think there is no doubts that a multiply modified protein will behave differently in the membrane, e.g. show different oligomerization properties and thus labeling (accessibility) efficiencies. In addition, the labeling efficiency or epitope accessibility depends on the labeling protocol, i.e. labeling before or after fixation (impacted also by the choice of fixative). Furthermore, it is known that the labeling efficiency is different for the basal and apical side on adherent cells. Finally, the labeling of nanobodies and antibodies with dyes and oligonucleotides can change their binding efficiency, as does the degree of labeling. Therefore, I do not believe that the method is generally applicable to determine the absolute labeling efficiency of a new antibody or nanobody clones.

Reviewer #2 (Remarks to the Author):

Second review of "Quantification of absolute labeling efficiency at the single-protein level" by Hellmeier et al.

In the revised manuscript, the authors have clarified the details of their analysis procedures, particularly with respect to the simulations which were used to determine the labeling efficiency. Supplementary Figure 1 provides a clear example of how the simulations work and why this is the right approach to obtain quantitative data. Also, by comparing their method with a previously published labeling efficiency estimation technique, the authors demonstrated a benchmark for the accuracy of their results.

I have one further suggestion, which is to include references to two earlier works, which presented ideas within a similar conceptual framework to the current study:

Wilmes, S. et al. Receptor dimerization dynamics as a regulatory valve for plasticity of type I interferon signaling. *J Cell Biol* 209, 579-593 (2015), and

Latty, S. L. et al. Referenced Single-Molecule Measurements Differentiate between GPCR Oligomerization States. *Biophys J* 109, 1798-1806 (2015).

My earlier comments were fully addressed in the revised manuscript. I consider this work to be innovative and immediately relevant to the fluorescence imaging field, and I recommend publication.

Reviewer #3 (Remarks to the Author):

The authors have addressed most of my concerns. I think the method will attract broad interest in the field. However, only demonstrating the technique based on the DNA PAINT imaging could quite limit the broad application of this method. Furthermore, the Supplementary Fig. 1 is missing.

Author Rebuttal, first revision:

Reviewers' Comments:

Reviewer #1:

Remarks to the Author:

I greatly appreciate the effort made by the authors to resolve my doubts about the general usefulness of the method. As already mentioned in my first assessment I do not question the results presented but I still disbelieve that the method will be valued and extensively used by the community. I think there is no doubts that a multiply modified protein will behave differently in the membrane, e.g. show different oligomerization properties and thus labeling (accessibility) efficiencies. In addition, the labeling efficiency or epitope accessibility depends on the labeling protocol, i.e. labeling before or after fixation (impacted also by the choice of fixative). Furthermore, it is known that the labeling efficiency is different for the basal and apical side on adherent cells. Finally, the labeling of nanobodies and antibodies with dyes and oligonucleotides can change their binding efficiency, as does the degree of labeling. Therefore, I do not believe that the method is generally applicable to determine the absolute labeling efficiency of a new antibody or nanobody clones.

We thank the reviewer for acknowledging the improved revised version of our study. We do believe that the concerns raised by the reviewer are actually features of our implementation and are by no means

limitations to the general applicability. To this end, we note, that it is important to conduct our workflow for evaluating labeling probe efficiency under the same cell fixation and target staining conditions that will be applied in subsequent experiments, where these probes are employed for protein target quantification. This is paramount as differences in fixation, permeabilization, and staining protocols will likely affect the resulting absolute probe labeling efficiency. Our assay can be customized to match specific experimental conditions, allowing for the direct benchmarking of probe labeling efficiency in the real-world conditions of downstream experiments.

Reviewer #2:

Remarks to the Author:

In the revised manuscript, the authors have clarified the details of their analysis procedures, particularly with respect to the simulations which were used to determine the labeling efficiency. Also Supplementary Figure 1 provides a clear example of how the simulations work and why this is the right approach to obtain quantitative data., by comparing their method with a previously published labeling efficiency estimation technique, the authors demonstrated a benchmark for the accuracy of their results.

I have one further suggestion, which is to include references to two earlier works, which presented ideas within a similar conceptual framework to the current study:

- Wilmes, S. et al. Receptor dimerization dynamics as a regulatory valve for plasticity of type I interferon signaling. *J Cell Biol* 209, 579-593 (2015), and
- Latty, S. L. et al. Referenced Single-Molecule Measurements Differentiate between GPCR Oligomerization States. *Biophys J* 109, 1798-1806 (2015).

My earlier comments were fully addressed in the revised manuscript. I consider this work to be innovative and immediately relevant to the fluorescence imaging field, and I recommend publication.

We are very grateful for the positive review and support of this referee. We are happy to include the suggested references and are now citing them in the introduction.

Reviewer #3:

Remarks to the Author:

The authors have addressed most of my concerns. I think the method will attract broad interest in the field. However, only demonstrating the technique based on the DNA PAINT imaging could quite limit the broad application of this method. Furthermore, the Supplementary Fig. 1 is missing.

We thank the reviewer for the supportive review of our work. We are grateful for the suggestion to add more information on the general applicability of our method beyond DNA-PAINT microscopy. From a conceptual standpoint of view, any imaging modality that is able to achieve single-target resolution (e.g. is able to distinguish single proteins in dense arrangements) and sensitivity can be used in our approach. So, if these requirements are reached, methods such as STORM, PALM, STED or any other super-resolution modality can be used. We now discuss this point more explicitly in the manuscript and provide guidance to the readers with regards to the requirements of applying the labeling efficiency evaluation workflow beyond DNA-PAINT microscopy. Supplementary Fig. 1 is now added, we apologize for this oversight in the original resubmission.

Final Decision Letter:

Dear Ralf,

I am pleased to inform you that your Article, "Quantification of absolute labeling efficiency at the single-protein level", has now been accepted for publication in Nature Methods. The received and accepted dates will be July 6, 2023 and March 11, 2024. This note is intended to let you know what to expect from us over the next month or so, and to let you know where to address any further questions.

Over the next few weeks, your paper will be copyedited to ensure that it conforms to Nature Methods style. Once your paper is typeset, you will receive an email with a link to choose the appropriate publishing options for your paper and our Author Services team will be in touch regarding any additional information that may be required. It is extremely important that you let us know now whether you will be difficult to contact over the next month. If this is the case, we ask that you send us the contact information (email, phone and fax) of someone who will be able to check the proofs and deal with any last-minute problems.

Please note that *Nature Methods* is a Transformative Journal (TJ). Authors may publish their research with us through the traditional subscription access route or make their paper immediately open access

through payment of an article-processing charge (APC). Authors will not be required to make a final decision about access to their article until it has been accepted. Find out more about Transformative Journals

If you are active on Twitter/X, please e-mail me your and your coauthors' handles so that we may tag you when the paper is published.

Best regards,
Rita

Rita Strack, Ph.D.
Senior Editor
Nature Methods